# Sparse Relaxed-Lasso Steering: Automatic Sparse Autoencoder Feature Selection for Precise Image Editing

**Zongxin Liu** [1 2]  **Xiaoyong Xue** [3]  **Weidi Sun** [3]  **Shengchao Qin** [4]  **Lijun Zhang** [1 2 5]

## Abstract

Precise, training-free image editing with text-to-image diffusion models requires balancing alignment (faithful realization of the target attribute), consistency (preserving non-target content), and quality (maintaining sharp, artifact-free textures). Sparse autoencoder (SAE) steering offers interpretable, smooth "slider-like" control by manipulating SAE feature activations derived from the text encoder; however, existing approaches rely on heuristic feature selection and manual steering-strength tuning, leading to suboptimal trade-offs among the three objectives. We propose Sparse Relaxed-Lasso Steering (SRLS), which casts steering-vector discovery as a convex sparse recovery problem. Exploiting the affine structure of the SAE decoder, SRLS automatically identifies sparse, generalizable support sets via a Lasso objective and then debiases the coefficients using support-restricted ridge refitting. We further replace manual strength tuning with a fixed-budget Bayesian optimization procedure. Across diverse attributes and subjects, SRLS improves the alignment–consistency–quality trade-off over competing methods.

## 1. Introduction

Text-to-image diffusion models can synthesize highly realistic images from natural-language prompts (Ho et al., 2020; Song et al., 2021; Peebles & Xie, 2023). As generation from scratch becomes routine, precise editing becomes increasingly important: given an image generated from a base

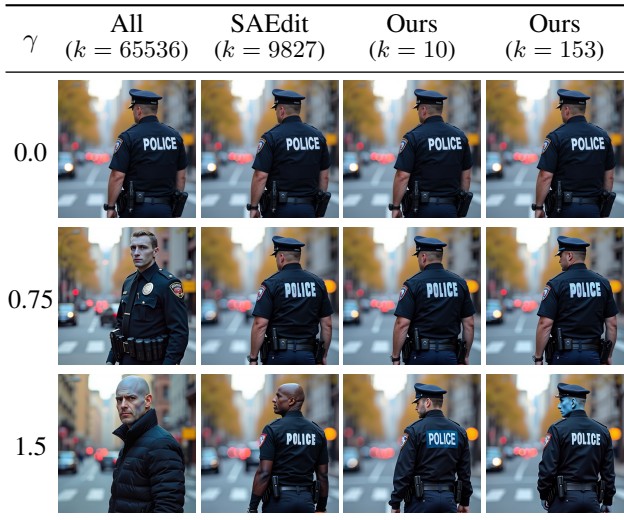

| $\gamma$ | All (k = 65536) | SAEdit (k = 9827) | Ours (k = 10) | Ours (k = 153) |
|---|---|---|---|---|
| 0.0 | | | | |
| 0.75 | | | | |
| 1.5 | | | | |

*Figure 1.* Editing results for adding the "blue skin" attribute to the base prompt "A police officer patrolling the street" with varying feature budgets $(k)$ and steering strengths $(\gamma)$. "All" uses all SAE features. SAEdit uses heuristic feature selection; SRLS $(k = 153)$ automatically selects suitable features; SRLS $(k = 10)$ enforces a maximum of 10 features.

prompt, we aim to inject a target attribute while preserving the original identity, geometry, and background.

Precise editing requires balancing three competing objectives: **alignment** (faithfully realizing the target attribute), **consistency** (preserving non-target content), and **quality** (maintaining sharp, artifact-free textures). Existing methods often struggle with this trade-off. Training-based approaches (Brooks et al., 2023; Zhang et al., 2023) offer strong controllability but incur substantial computational cost and reduce flexibility for arbitrary concepts. Training-free approaches (Meng et al., 2022; Hertz et al., 2023; Chefer et al., 2023; Cao et al., 2023; Tumanyan et al., 2023; Parmar et al., 2023; Brack et al., 2023) manipulate attention or latent representations in frozen models, but can be brittle: they may induce geometric drift and texture degradation, or fail to manifest the desired attribute, while prompt regeneration often changes unrelated scene elements (Figure 2).

Recent progress in mechanistic interpretability, particularly sparse autoencoders (SAEs) (Kamenetsky et al., 2025; Kim

---

[1]Key Laboratory of System Software (Chinese Academy of Sciences), Institute of Software Chinese Academy of Sciences, Beijing, China [2]University of Chinese Academy of Sciences, Beijing, China [3]School of Mathematical Sciences, Peking University, Beijing, China [4]Xidian University, Xi'an, China [5]Institute of AI for Industries, Chinese Academy of Sciences, Nanjing, Jiangsu, China. Correspondence to: Lijun Zhang <zhanglj@ios.ac.cn>.

*Proceedings of the 43rd International Conference on Machine Learning*, Seoul, South Korea. PMLR 306, 2026. Copyright 2026 by the author(s).

| Prompts | Base | Ours | SAEdit | SEGA | Pix2Pix-Zero | Prompt Only |
|---------|------|------|--------|------|--------------|-------------|
| A large tiger (with **green skin**) walking in the jungle | | | | | | |
| A photo of a man (with a **long beard**) standing in the room | | | | | | |
| A cute rabbit (**made of stone**) on the grass | | | | | | |

*Figure 2.* Qualitative comparison of training-free diffusion-based editing methods. "Base" is generated without the parenthetical attribute, whereas "Prompt Only" regenerates the image using the full prompt. Bold text indicates the target attribute.

& Ghadiyaram, 2025), provides a promising route to disentangled and interpretable control. SAEs map dense activations into an overcomplete sparse feature space, exposing concept-related features that can be manipulated for editing. However, SAE-based editing methods such as SAEdit rely on heuristic feature selection (e.g., PCA over top-$k$ activated features). In practice, **feature selection** and **steering strength** are both critical: selecting too many features ("All") or less suitable features (SAEdit) can degrade image quality, whereas selecting too few yields ineffective edits; moreover, tuning the steering strength $\gamma$ is essential to balance attribute alignment and content preservation (Figure 1).

To address these challenges, we propose **Sparse Relaxed-Lasso Steering (SRLS)**, a framework that formulates steering-vector discovery as a sparse signal recovery problem in the SAE feature space. Exploiting the affine structure of SAE decoders, we reduce steering discovery to a convex Lasso objective that matches the target reconstruction while enforcing sparsity, yielding compact supports that empirically transfer across prompts. Our contributions are:

(1) We formulate feature selection as a sparse optimization problem, use a $\lambda$-path search to identify the feature support automatically, and apply support-restricted ridge refitting to debias Lasso shrinkage.

(2) We use Bayesian optimization to select the steering strength $\gamma$ automatically, avoiding manual tuning.

(3) Experiments show that SRLS outperforms competitive baselines, achieving a better balance among alignment, consistency, and quality.

## 2. Related Work

Building on DDPM/DDIM (Ho et al., 2020; Song et al., 2021), text-to-image diffusion has progressed from pixel-space U-Nets to latent diffusion with cross-attention (Rombach et al., 2022), and further to Diffusion Transformers (DiT) with ViT-style latent denoisers (Peebles & Xie, 2023).

Training-based methods improve controllability via extra training or adapters, e.g., instruction-following edits (InstructPix2Pix (Brooks et al., 2023)) and structure-conditioned branches (ControlNet (Zhang et al., 2023)), as well as personalization (Textual Inversion (Gal et al., 2023), DreamBooth (Ruiz et al., 2023)). They are effective but require additional data, computation, and model management.

Training-free methods edit frozen models at inference time by intervening in the sampling process or conditioning signals. SDEdit (Meng et al., 2022) perturbs the input with noise and denoises it, using the noise level to trade off faithfulness and realism. Prompt-to-Prompt (Hertz et al., 2023) manipulates cross-attention maps to preserve layout under prompt edits, while Attend-and-Excite (Chefer et al., 2023) increases attention to selected tokens to improve prompt adherence. SEGA (Brack et al., 2023) composes semantic guidance directions under classifier-free guidance (CFG), and Pix2Pix-Zero (Parmar et al., 2023) estimates edit directions in text embedding space with attention based guidance.

Recent SAE-based control methods leverage sparse, interpretable feature spaces to derive semantic steering directions by intervening on text-encoder activations/embeddings.

SAEdit (Kamenetsky et al., 2025) selects a steering direction using top-$k$ activated features and PCA, while Concept Steerers (Kim & Ghadiyaram, 2025) directly use top-$k$ features for concept-level control (e.g., unsafe content removal). However, both rely on heuristic feature selection and manual strength tuning, which, in some cases, may lead to suboptimal trade-offs between alignment, consistency, and quality.

## 3. Preliminaries

This section reviews background on text-to-image diffusion models and sparse autoencoders.

### 3.1. Text-to-Image Diffusion Model Overview

We consider DiT-based text-to-image models with three components: (1) A text encoder $f_{\text{enc}}(\cdot)$ mapping an $L$-token prompt $x$ to token embeddings $\mathbf{H} = f_{\text{enc}}(x) \in \mathbb{R}^{m \times L}$, where $\mathbf{h}_i \in \mathbb{R}^m$ is the $i$-th token embedding, (2) A DiT denoising backbone that iteratively denoises in latent space while attending to text via cross-attention, (3) A VAE decoder that maps the final latent to an RGB image. We freeze all model components and intervene only on the text-conditioning representations.

### 3.2. Sparse Autoencoder on Text-Encoder Activations

Sparse autoencoders (SAEs) are used in mechanistic interpretability under the superposition view: many features share an activation space and behave as approximately linear directions, so activations can be decomposed using a sparse dictionary. Ideally, each feature direction matches a human-understandable concept.

Given an embedding sequence $\mathbf{H} = [\mathbf{h}_1, \ldots, \mathbf{h}_L] \in \mathbb{R}^{m \times L}$, an SAE consists of an encoder $E : \mathbb{R}^m \to \mathbb{R}^n$ and a decoder $D : \mathbb{R}^n \to \mathbb{R}^m$ with $n \gg m$. We extend $E$ and $D$ to act column-wise:

$$E(\mathbf{H}) = [E(\mathbf{h}_1), \ldots, E(\mathbf{h}_L)]$$
$$= [\mathbf{s}_1, \ldots, \mathbf{s}_L] =: \mathbf{S} \in \mathbb{R}^{n \times L}.$$

In many SAE implementations for interpretability, the decoder is affine, i.e.,

$$D(\mathbf{s}) = \mathbf{W}_d \mathbf{s} + \mathbf{b}_d.$$

Accordingly, for $\mathbf{S} \in \mathbb{R}^{n \times L}$, we have:

$$D(\mathbf{S}) = [D(\mathbf{s}_1), \ldots, D(\mathbf{s}_L)] = \mathbf{W}_d \mathbf{S} + \mathbf{b}_d \mathbf{1}_L^\top \in \mathbb{R}^{m \times L}.$$

**Steering via SAE.** Given a learned steering vector $\Delta \mathbf{s} \in \mathbb{R}^n$ and a steering strength $\gamma$, we modify SAE latents before decoding. Steering can target a subset of tokens indexed by $\mathcal{T} \subseteq \{1, 2, \ldots, L\}$:

$$\mathbf{S}' = \mathbf{S} + \gamma \Delta \mathbf{s} \mathbf{a}^\top, \quad [\mathbf{a}]_i = \mathbb{I}[i \in \mathcal{T}], \tag{1}$$

where $\mathbb{I}[\cdot]$ is the indicator function. Here, $\mathbf{a} \in \{0, 1\}^L$ indicates the selected tokens, and $\mathbf{S}' \in \mathbb{R}^{n \times L}$ denotes the steered latent sequence. Eq. (1) uses a constant steering strength for notation; at inference time, $\gamma$ is replaced by a denoising-step-dependent strength. The steered embeddings are then decoded token-wise:

$$\widehat{\mathbf{h}}_i = D(\mathbf{s}_i'), \quad \widehat{\mathbf{H}} = \left[ \widehat{\mathbf{h}}_1, \widehat{\mathbf{h}}_2, \ldots, \widehat{\mathbf{h}}_L \right] \in \mathbb{R}^{m \times L}. \tag{2}$$

In our setup, $\widehat{\mathbf{H}}$ is then used as the text-conditioning input to the frozen diffusion backbone. Our goal is to learn, from contrastive prompt pairs, a general $\Delta \mathbf{s}$ for a target concept that can be applied to arbitrary prompts at inference.

## 4. Sparse Relaxed-Lasso Steering

In this section, we formulate steering-vector discovery as a convex sparse signal recovery problem in the SAE feature space. Figure 3 and Algorithm 1 illustrate the overall pipeline. We aggregate token-wise SAE features into prompt-level targets, optimize a Lasso objective in the SAE decoder space to learn a sparse steering vector, and then use a $\lambda$-path search with support-restricted ridge refitting to control sparsity and debias coefficients.

### 4.1. Problem Formulation and Token Aggregation

Given $N$ contrastive prompt pairs $\{(x_i^+, x_i^-)\}_{i=1}^N$, where $x_i^+$ includes the target concept and $x_i^-$ does not, let $\mathbf{S}_i^+ \in \mathbb{R}^{n \times L_i^+}$ and $\mathbf{S}_i^- \in \mathbb{R}^{n \times L_i^-}$ denote the corresponding token-wise SAE latent sequences.

To map token-level activations to a prompt-level representation, we aggregate across tokens with element-wise pooling: $\bar{\mathbf{s}}_i^+ \triangleq A(\mathbf{S}_i^+)$, $\bar{\mathbf{s}}_i^- \triangleq A(\mathbf{S}_i^-)$, where $A$ is the aggregation function (e.g., max or mean pooling) along the token axis. We use mean pooling to capture overall semantics. The aggregation $A(\cdot)$ is used only to build a stable prompt-level target for learning a universal direction $\Delta \mathbf{s}$. At inference time, we apply $\Delta \mathbf{s}$ to the token sequence as described in Equation (1).

### 4.2. Reconstruction-Space Alignment Objective

Our core insight is that if a global editing direction exists, then adding this direction to the sparse representation of a negative prompt should produce a decoded output that closely matches the decoded positive prompt.

We therefore seek a global sparse steering vector that minimizes the reconstruction difference between the steered negative representation and the positive target. For a regularization strength $\lambda \geq 0$, we consider the following sparse reconstruction objective:

$$\min_{\Delta \mathbf{s} \in \mathbb{R}^n} \mathcal{L}_{\text{rec}}(\Delta \mathbf{s}) + \lambda \|\Delta \mathbf{s}\|_1, \tag{3}$$

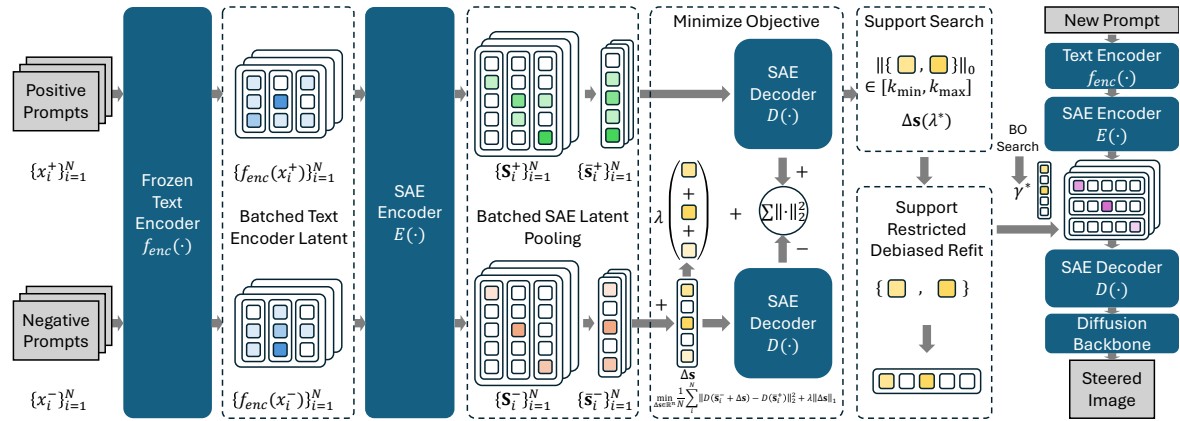

*Figure 3.* Overview of the SRLS pipeline.

where the reconstruction loss is defined as:

$$\mathcal{L}_{\text{rec}}(\Delta \mathbf{s}) = \frac{1}{N} \sum_{i=1}^{N} \left\| D(\bar{\mathbf{s}}_i^- + \Delta \mathbf{s}) - D(\bar{\mathbf{s}}_i^+) \right\|_2^2. \quad (4)$$

This objective learns a sparse direction whose decoded effect matches the average attribute shift across the contrastive prompt pairs. We align in the SAE reconstruction space because differences in the original embeddings can include components outside the range of $\mathbf{W}_d$, which makes some target directions unattainable and can lead to dense, unstable solutions. In reconstruction space, the differences are realizable, and the affine decoder reduces the problem to a convex Lasso objective, enabling sparse support discovery and more reliable control.

### 4.3. Exact Objective Reduction

In standard SAE implementations, the decoder is affine: $D(\mathbf{s}) = \mathbf{W}_d \mathbf{s} + \mathbf{b}_d$. Substituting this into the objective, the bias term $\mathbf{b}_d$ cancels out:

$$\mathcal{L}_{\text{rec}}(\Delta \mathbf{s}) = \frac{1}{N} \sum_{i=1}^{N} \left\| \mathbf{W}_d \Delta \mathbf{s} - \mathbf{d}_i \right\|_2^2, \quad (5)$$

where $\mathbf{d}_i \triangleq \mathbf{W}_d(\bar{\mathbf{s}}_i^+ - \bar{\mathbf{s}}_i^-)$.

**Proposition 1 (Exact Reduction to Mean Target).** Let $\bar{\mathbf{d}} = \frac{1}{N} \sum_{i=1}^{N} \mathbf{d}_i$ be the mean target difference in the decoder output space. For any fixed $\lambda \geq 0$, the optimization problem in Equation (3) is equivalent to the Lasso objective:

$$\Delta \mathbf{s}(\lambda) \in \arg \min_{\Delta \mathbf{s} \in \mathbb{R}^n} \left\| \mathbf{W}_d \Delta \mathbf{s} - \bar{\mathbf{d}} \right\|_2^2 + \lambda \|\Delta \mathbf{s}\|_1. \quad (6)$$

*Proof.* The sum of squared errors can be decomposed as

$$\sum_{i=1}^{N} \|\mathbf{W}_d \Delta \mathbf{s} - \mathbf{d}_i\|_2^2 = N\|\mathbf{W}_d \Delta \mathbf{s} - \bar{\mathbf{d}}\|_2^2 + \sum_{i=1}^{N} \|\mathbf{d}_i - \bar{\mathbf{d}}\|_2^2.$$

Dividing both sides by $N$ and discarding the constant term yields the fixed-$\lambda$ Lasso objective in Equation (6). $\square$

### 4.4. Adaptive Sparsity Control and Debiased Refitting

To keep the steering vector interpretable and effective, we use sparsity control and support-restricted debiasing.

**$\lambda$-Path Search for a Target Support Size.** The solutions $\{\Delta \mathbf{s}(\lambda) : \lambda \geq 0\}$ of Equation (6) define a Lasso solution path, along which the active support varies with the regularization strength. We target a support size $\|\Delta \mathbf{s}(\lambda)\|_0 \in [k_{\min}, k_{\max}]$ that balances expressiveness and disentanglement. The range $[k_{\min}, k_{\max}]$ serves mainly as a safeguard against degenerate solutions that are either too sparse to express the target concept or too dense to remain disentangled.

We search over $\lambda$, solving Eq. (6) via ISTA at each step $p$:

$$\mathbf{u}^{(p)} = \Delta \mathbf{s}^{(p)} - 2\eta \mathbf{W}_d^\top \left( \mathbf{W}_d \Delta \mathbf{s}^{(p)} - \bar{\mathbf{d}} \right), \quad (7)$$

$$\Delta \mathbf{s}^{(p+1)} = \text{ST}_{\eta\lambda}(\mathbf{u}^{(p)}), \quad (8)$$

where $\text{ST}_{\eta\lambda}(z) = \text{sign}(z) \cdot \max(|z| - \eta\lambda, 0)$ is the soft-thresholding operator, applied element-wise, and $\eta$ is the step size. We define the target regularization parameter by the continuous selection criterion

$$\lambda^\star \in \arg \min_{\lambda : \|\Delta \mathbf{s}(\lambda)\|_0 \in [k_{\min}, k_{\max}]} \left\| \mathbf{W}_d \Delta \mathbf{s}(\lambda) - \bar{\mathbf{d}} \right\|_2^2. \quad (9)$$

When multiple minimizers exist for a given $\lambda$, $\Delta \mathbf{s}(\lambda)$ denotes the solution returned by the numerical solver, and $\mathcal{J}$ is defined as the support of that selected solution. In practice, Equation (9) is approximated by a bisection-style search over $\lambda \in [\lambda_{\min}, \lambda_{\max}]$. Since the Lasso active set need not be strictly monotone for correlated dictionaries, the search is heuristic; we track the feasible solutions encountered during the search and return the one with the smallest reconstruction error. For ISTA, we use a fixed step size $\eta = 1/C_{\text{Lip}}$, where $C_{\text{Lip}} = 2\|\mathbf{W}_d\|_2^2$ is the Lipschitz constant of the gradient, and we stop when the relative change in $\Delta \mathbf{s}$ is below a threshold or after a fixed number of iterations.

**Support-Restricted Ridge Refitting.** The $\ell_1$ penalty shrinks coefficients. To reduce Lasso-induced shrinkage

**Algorithm 1** SRLS with Support-Restricted Ridge Refit

**Require:** Pairs $\{(x_i^+, x_i^-)\}_{i=1}^N$; text encoder $f_{\text{enc}}$; SAE $(E, D)$ with affine decoder matrix $\mathbf{W}_d$; target support-size range $[k_{\min}, k_{\max}]$; ridge penalty $\rho$.

**Ensure:** Debiased sparse steering vector $\Delta \mathbf{s}^\star$.

1: **Step 1: Token aggregation**
2: **for** $i = 1$ to $N$ **do**
3:    $\mathbf{S}_i^+ \leftarrow E(f_{\text{enc}}(x_i^+)), \quad \mathbf{S}_i^- \leftarrow E(f_{\text{enc}}(x_i^-))$
4:    $\bar{\mathbf{s}}_i^+ \leftarrow A(\mathbf{S}_i^+), \quad \bar{\mathbf{s}}_i^- \leftarrow A(\mathbf{S}_i^-)$
5: **end for**
6: $\bar{\mathbf{d}} \leftarrow \frac{1}{N} \sum_{i=1}^N \mathbf{W}_d(\bar{\mathbf{s}}_i^+ - \bar{\mathbf{s}}_i^-)$
7: **Step 2: $\lambda$-path search for target support size**
8: Select $\lambda^\star$ approximately by Equation (9), using a bisection-style search over Equation (6).
9: $\mathcal{J} \leftarrow \operatorname{supp}(\Delta \mathbf{s}(\lambda^\star))$
10: **Step 3: Support-restricted ridge refit**
11: $\mathbf{W}_{d,\mathcal{J}} \leftarrow$ columns of $\mathbf{W}_d$ indexed by $\mathcal{J}$
12: $\boldsymbol{\alpha}_{\mathcal{J}}^\star \leftarrow (\mathbf{W}_{d,\mathcal{J}}^\top \mathbf{W}_{d,\mathcal{J}} + \rho \mathbf{I}_{|\mathcal{J}|})^{-1} \mathbf{W}_{d,\mathcal{J}}^\top \bar{\mathbf{d}}$
13: Construct $\Delta \mathbf{s}^\star$ from $\boldsymbol{\alpha}_{\mathcal{J}}^\star$ on support $\mathcal{J}$ and zeros elsewhere.
14: **return** $\Delta \mathbf{s}^\star$

while preserving the selected support, we run a ridge-regularized refit restricted to the active support:

$$\mathcal{J} = \operatorname{supp}(\Delta \mathbf{s}(\lambda^\star)). \tag{10}$$

Let $\mathbf{W}_{d,\mathcal{J}}$ denote the submatrix of $\mathbf{W}_d$ formed by columns indexed by $\mathcal{J}$. We compute

$$\boldsymbol{\alpha}_{\mathcal{J}}^\star = (\mathbf{W}_{d,\mathcal{J}}^\top \mathbf{W}_{d,\mathcal{J}} + \rho \mathbf{I}_{|\mathcal{J}|})^{-1} \mathbf{W}_{d,\mathcal{J}}^\top \bar{\mathbf{d}}, \tag{11}$$

where $\rho > 0$ is a small ridge penalty to stabilize the inversion, and $\mathbf{I}_{|\mathcal{J}|}$ is the identity matrix of size $|\mathcal{J}|$; in practice $\rho$ can be fixed across concepts (e.g., $10^{-4}$) or set proportional to $\operatorname{trace}(\mathbf{W}_{d,\mathcal{J}}^\top \mathbf{W}_{d,\mathcal{J}})/|\mathcal{J}|$. The final, debiased steering vector $\Delta \mathbf{s}^\star$ is then constructed by assigning $\boldsymbol{\alpha}_{\mathcal{J}}^\star$ to indices in $\mathcal{J}$ and setting all remaining coordinates to zero.

### 4.5. Inference-Time Editing Protocol

Following SAEdit (Kamenetsky et al., 2025), we use the same denoising-time steering schedule and hyperparameter settings. Let $t \in \{0, \ldots, T-1\}$ index steps from early structure formation to later detail refinement, and let $q_t = \frac{t}{T-1} \in [0, 1]$. We set

$$\Gamma_t(\gamma) = \operatorname{sign}(\gamma) \min \left( \exp(q_t |\gamma|) - 1, \tau \right), \tag{12}$$

where $\gamma$ is the base strength and $\tau$ is the clipping threshold, for which we use the same value as SAEdit in all experiments. The schedule keeps early interventions weak and increases them during later refinement, improving attribute injection while preserving layout. We use it only for inference-time modulation; our method differs from SAEdit in how the sparse direction is selected and how $\gamma$ is chosen.

## 5. Automatic Steering-Strength Selection

Given the final sparse steering vector $\Delta \mathbf{s}^\star$ returned by Algorithm 1, the remaining inference-time hyperparameter is the scalar base strength $\gamma$ in Equation (12). Note that strength selection only searches for a value of $\gamma$ that balances target alignment, content preservation, and image quality.

For an input prompt $x^-$ and a target prompt $x^+$, with text-encoder activations $\mathbf{H} = f_{\text{enc}}(x^-) \in \mathbb{R}^{m \times L}$, let $\mathbf{S} = E(\mathbf{H}) \in \mathbb{R}^{n \times L}$ denote the corresponding SAE latent sequence. We fix a token set $\mathcal{T} \subseteq \{1, \ldots, L\}$ before strength selection, and define the binary token mask

$$[\mathbf{a}]_i = \mathbb{I}[i \in \mathcal{T}], \qquad \mathbf{a} \in \{0, 1\}^L. \tag{13}$$

At denoising step $t$, the scalar base strength $\gamma$ is converted into an applied strength through the schedule $\Gamma_t(\gamma)$ in Equation (12). The steered SAE latents are then

$$\mathbf{S}_t'(\gamma) = \mathbf{S} + \Gamma_t(\gamma) \Delta \mathbf{s}^\star \mathbf{a}^\top, \tag{14}$$

and the corresponding steered text-conditioning sequence is

$$\widehat{\mathbf{H}}_t(\gamma; x^-) = D(\mathbf{S}_t'(\gamma)) \in \mathbb{R}^{m \times L}. \tag{15}$$

Thus, $\Delta \mathbf{s}^\star$ and $\mathbf{a}$ determine where and along which SAE direction to intervene, while $\gamma$ controls the overall strength.

Let $\mathcal{G}$ denote the full generation pipeline when the random seed is fixed, including the frozen diffusion backbone and the VAE decoder. Given the scheduled conditioning sequence $\{\widehat{\mathbf{H}}_t(\gamma; x^-)\}_t$, the generated image is

$$\mathbf{I}(\gamma; x^-) := \mathcal{G}\left(\{\widehat{\mathbf{H}}_t(\gamma; x^-)\}_t\right). \tag{16}$$

We evaluate the generated image with a reward $\mathcal{R}$ measuring target alignment, content consistency, and image quality. In our experiments, $\mathcal{R}$ is the harmonic mean of LLM-based Alignment, Consistency, and Quality scores, rewarding better balance among the three aspects while penalizing degenerate solutions that achieve high scores on some aspects at the cost of others.

The automatic strength selection problem can therefore be formulated as the following one-dimensional black-box optimization:

$$\begin{aligned} \gamma^\star &= \arg \max_{\gamma \in [\gamma_{\min}, \gamma_{\max}]} r(\gamma), \\ r(\gamma) &= \mathcal{R}\left(\mathbf{I}(\gamma; x^-); x^+, x^-, \mathbf{I}(0; x^-)\right). \end{aligned} \tag{17}$$

Note that only the scalar base strength $\gamma$ is optimized; the token mask $\mathbf{a}$, steering vector $\Delta \mathbf{s}^\star$, schedule, and clipping threshold $\tau$ are fixed.

Because $r(\gamma)$ depends on full diffusion sampling and may involve non-differentiable image-quality or vision-language judgments, we treat it as a black-box objective. We solve

Equation (17) with Bayesian optimization using a Gaussian-process surrogate with an RBF kernel and an expected-improvement acquisition function. The one-dimensional search is efficient and avoids manual tuning for each prompt–attribute pair.

# 6. Experiments

We organize experiments around three questions:

(1) **RQ1** How effective are SRLS's feature selection strategies for image editing?

(2) **RQ2** How effective is SRLS's automated strength-selection procedure?

(3) **RQ3** How well does our method generalize across subjects and editing scenarios?

## 6.1. Experimental Setup

**Model and compute.** We mainly evaluate on FLUX.1-dev (Black Forest Labs, 2024) with a pre-trained SAE on the final layer of the T5 text encoder, following SAEdit (Kamenetsky et al., 2025). Appendix E reports Stable Diffusion 3.5 Large results (Esser et al., 2024). All experiments use a 40 GB NVIDIA A100 GPU and random seed 42 unless otherwise specified.

**Baselines and implementation details.** We compare our method with the following baselines:

(1) **SAEdit** (Kamenetsky et al., 2025) using the authors' open-source implementation and recommended hyper-parameters;

(2) **All**, which uses the mean difference between positive and negative prompt pairs in SAE sparse space as the steering direction without feature selection;

(3) **SEGA** (Brack et al., 2023) and **Pix2Pix-Zero** (Parmar et al., 2023), adapted to FLUX.1-dev with recommended hyperparameters;

(4) **Prompt Only**, which generates images directly from the full prompt with the target attribute rather than editing a base image.

We implement SRLS with a support range of $[5, 500]$, 200 ISTA iterations per $\lambda$, and ridge regularization $\rho = 10^{-4}$. For SAE-based methods, we normalize each steering vector to unit $\ell_2$ norm for comparable strengths.

**Editing tasks.** We evaluate 33 attributes: 14 concepts with positive/negative prompt pairs from SAEdit (Kamenetsky et al., 2025), plus 19 complex attributes (e.g., "made of stone", "blue skin") generated with the templates in Appendix K. Each attribute has 50–150 prompt pairs. Generalization is tested on 17 unseen base prompts spanning animals, people, professions, and fictional characters. We compare steering all tokens with steering manually selected

subject tokens only (e.g., "rabbit" in "a cute rabbit").

**Sampling protocol and strength search.** For each prompt–attribute–token combination, we generate 21 edited images with $\gamma \in \{1, 1.5, \ldots, 11\}$. For the three SAE-based methods, this yields 70,686 images. For Bayesian optimization (BO), we run 21 iterations over $\gamma \in [1, 11]$ to match the grid budget. All candidate images in both cases are scored by the mean of three independent evaluations to reduce noise.

## 6.2. Evaluation Metrics

**LLM-based evaluation.** We use an LLM to score Consistency, Quality, and Alignment on a 1–5 scale. Appendix L gives the rubrics. For reproducibility and cost control, we use Qwen3-VL-30B-A3B-Thinking (Bai et al., 2025), score each edited image three times, and report the average.

Because practical editing often involves trying multiple strengths and different methods produce vectors with different activation scales, we select, for each prompt–attribute–token combination, the strength that yields the highest harmonic mean of LLM scores. The harmonic mean treats consistency, quality, and alignment as equally important. Appendix B reports fixed-$\gamma$ results.

**Objective metrics.** To complement LLM scoring, we report `pyiqa` metrics (Chen & Mo, 2022). Let $\mathbf{I}_{\text{base}}$ be the base-prompt image, $\mathbf{I}_{\text{edit}}$ the edited image, and $x^+/x^-$ the target/base prompt. We use two metrics for each aspect:

(1) **Quality (no-reference, $\mathbf{I}_{\text{edit}}$):** TOPIQ-NR↑ (semantic-aware perceptual quality; higher is better) (Chen et al., 2024), and MUSIQ↑ (multi-scale Transformer for perceptual quality; higher is better) (Ke et al., 2021).

(2) **Consistency (reference-based, $\mathbf{I}_{\text{base}}, \mathbf{I}_{\text{edit}}$):** LPIPS↓ (deep-feature perceptual distance; lower values indicate better preservation of structure/identity/layout) (Zhang et al., 2018), and DISTS↓ (deep-feature full-reference distance capturing structure/texture with tolerance to texture resampling; lower is better) (Ding et al., 2022).

(3) **Alignment (image–text, $\mathbf{I}_{\text{edit}}, \mathbf{I}_{\text{base}}, x^+, x^-$):** CLIP↑ (cosine similarity between CLIP image/text embeddings for matching $\mathbf{I}_{\text{edit}}$ to $x^+$) (Radford et al., 2021), and DIR-CLIP↑ (alignment between image edit direction $\Delta \mathbf{I} = f_I(\mathbf{I}_{\text{edit}}) - f_I(\mathbf{I}_{\text{base}})$ and text direction $\Delta \mathbf{T} = f_T(x^+) - f_T(x^-)$) (Gal et al., 2022).

## 6.3. Effectiveness of Feature Selection (RQ1)

**LLM-based evaluation.** Figure 4 reports LLM-based results when steering all tokens. SRLS (without BO) ranks first or second across all metrics. Notably, SRLS achieves a harmonic mean of 3.97, surpassing SAEdit (3.89) and All (3.32). Our method also achieves the highest Pass(all) rates, indicating that SRLS can consistently produce high-

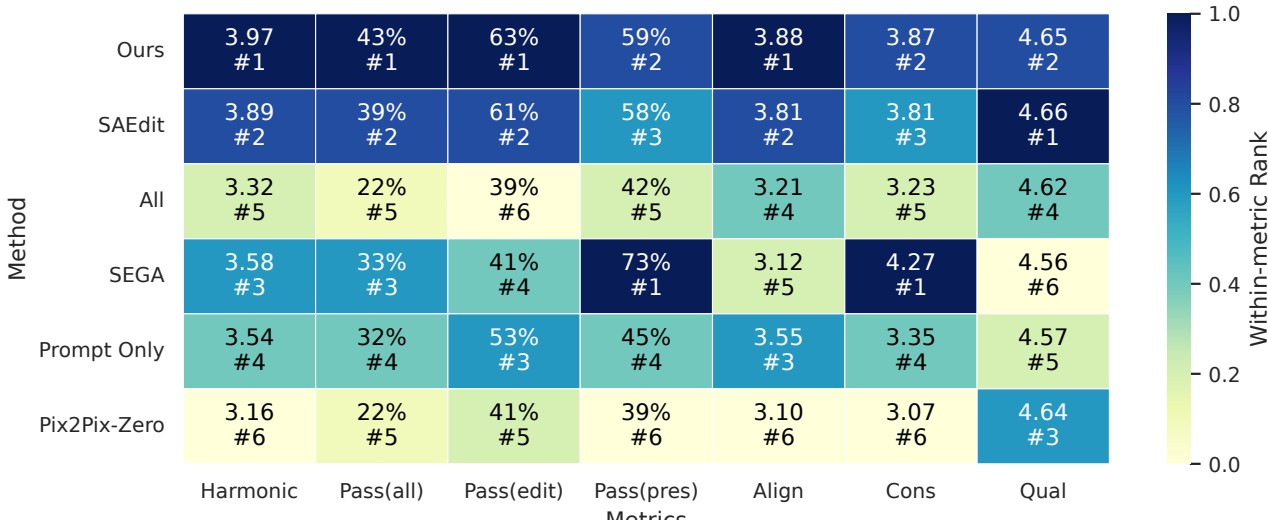

*Figure 4.* LLM evaluation for all-token steering. Align/Cons/Qual denote average Alignment/Consistency/Quality scores. Harmonic is computed per sample and then averaged across samples. Pass(pres) requires Cons≥4 and Qual≥4; Pass(edit) requires Align≥4; Pass(all) requires Cons≥4, Qual≥4, and Align≥4. Colors indicate method ranks; "#" gives the rank.

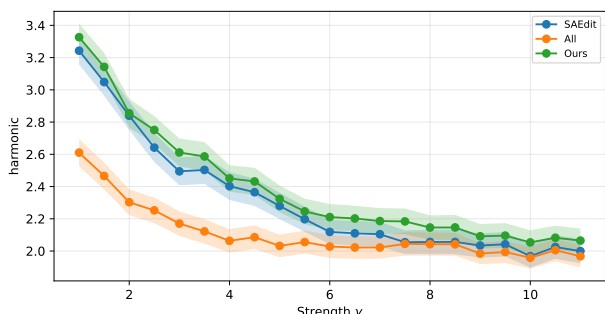

*Figure 5.* Fixed-strength robustness for all-token steering. Shaded areas indicate 95% confidence intervals.

*Table 1.* Mean number of selected features (Mean #Features) and harmonic mean (HM) of LLM scores by category.

| | Methods | All | SAEdit | Ours |
|---|---|---|---|---|
| | Mean #Features ($k$) | 65536 | 7859.15 | **307.39** |
| HM | Animals | 3.53 | 3.87 | **3.91** |
| | Humans | 3.03 | 3.73 | **3.80** |
| | Professions | 3.44 | 4.13 | **4.22** |
| | Fictional | 3.23 | 3.74 | **3.85** |

quality, well-aligned edits while preserving image consistency. Appendix A provides additional LLM results for subject-token steering, confirming that our method outperforms other SAE-based methods in most cases.

**Fixed-strength robustness.** The main LLM tables select the best steering strength for each prompt–attribute–token combination. To verify that the ranking is not an artifact of this protocol, Figure 5 reports the mean harmonic score over fixed steering strengths $\gamma$ when steering all tokens. SRLS stays above the SAE baselines across a broad range of strengths, showing that sparse feature selection improves the trade-off between attribute injection and image consistency even without per-example strength selection.

**Objective evaluation.** Figure 6 shows objective results for SAE-based methods. At fixed strengths, SRLS usually achieves higher image quality than SAEdit and All. For

consistency, SRLS is comparable to SAEdit and better than All at lower strengths, but degrades at high strengths as the exponential schedule becomes clipped; beyond that point, increasing $\gamma$ shifts additional changes to earlier denoising steps. These steps respond weakly to All but still respond to our selected features, so SRLS retains clear control effects at high strengths (Figure 7). For alignment, SRLS almost always surpasses the other two, indicating stronger attribute injection while maintaining quality and consistency.

**Qualitative comparison.** Figure 2 visualizes representative prompt–attribute pairs. SRLS inserts the target attribute while better preserving structure and details; other methods either under-edit or reduce consistency. For the green-skin tiger, SRLS changes skin color while retaining texture and background, whereas SAEdit alters pose and texture and other methods remain close to the base.

**Feature Sparsity and Generalization.** Table 1 shows that across all subjects, SRLS achieves higher harmonic mean scores with fewer features. Figure 8 further shows that a single steering vector learned by our method can add

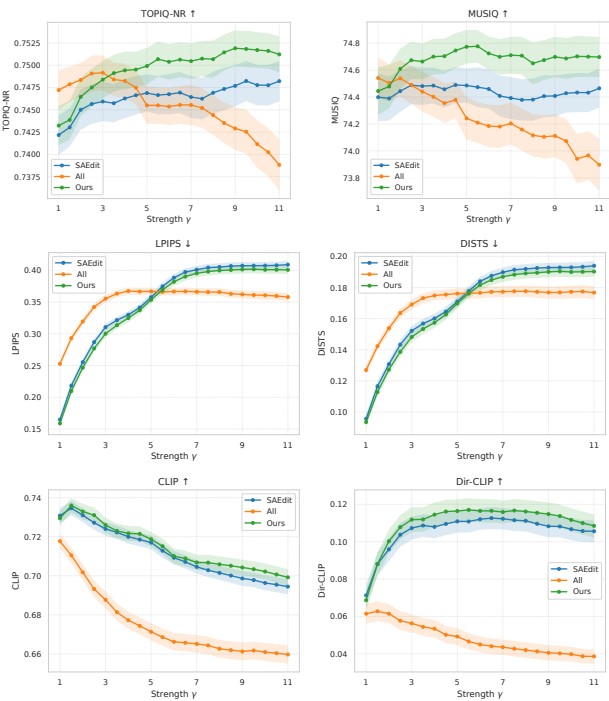

Figure 6. Objective evaluation for SAE-based methods. Scores are averaged over attributes and prompts at each strength; arrows show the better direction and shaded regions denote standard deviations.

"sunglasses" across animals, humans, fictional characters, and professions, whereas SAEdit fails for animal subjects. Appendix I provides qualitative examples demonstrating that SRLS selects interpretable, attribute-relevant features.

**Component ablations and sensitivity.** Under the reduced $\gamma$ sweep used in Appendix H, Lasso-only improves the harmonic mean from 3.08 (All) to 3.71, showing that sparse support discovery is the main driver. Support-restricted ridge refitting raises the score to 3.77 by mitigating shrinkage, and mean pooling outperforms max pooling for estimating the global attribute shift. Performance saturates around 50 contrastive pairs per attribute, and the default support range $[5, 500]$ behaves similarly to an unconstrained search, indicating our method can automatically choose a suitable support size in most cases.

> **Answer to RQ 1:** Empirical evaluations show that SRLS achieves better alignment–consistency trade-offs with far fewer features than prior methods. Stable Diffusion 3.5 Large results (Appendix E) confirm the finding, and the selected features generalize across subjects with strong detail preservation.

### 6.4. Effectiveness of Strength Selection (RQ2)

Table 2 compares grid search and Bayesian optimization (BO) as two test-time strength-selection strategies under

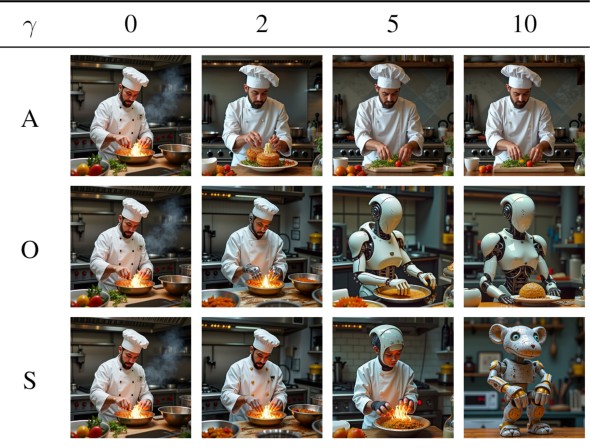

Figure 7. Adding the "cyborg" attribute to "A chef cooking in a kitchen" at different strengths. A: All; O: Ours; S: SAEdit. SRLS and SAEdit continue strengthening the edit, while All saturates.

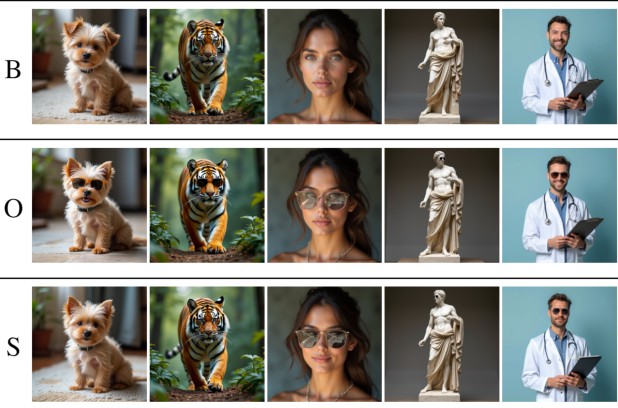

Figure 8. Generalization comparison for adding sunglasses with one steering vector across animals, humans, fictional characters, and professions. B: Base; O: Ours; S: SAEdit.

the same evaluation budget. The grid baseline evaluates 21 uniformly spaced strengths $\gamma \in \{1, 1.5, \ldots, 11\}$, whereas BO uses the same number of reward evaluations to search the continuous interval $[1, 11]$ with a Gaussian-process surrogate and an expected-improvement acquisition function. For both strategies, we report the score of the candidate selected by the LLM-based harmonic reward, so the comparison reflects strength-selection performance under a fixed evaluation budget.

Compared with the grid baseline, BO improves all three SAE-based methods. SRLS+BO achieves the best HM, Consistency, Alignment, and pass rates, while remaining close to the best Quality score. Since BO only changes the scalar strength $\gamma$, the improvement comes from selecting a better strength for the same steering direction rather than from changing the learned sparse vector. This suggests that BO strength selection complements sparse feature selection.

*Table 2.* BO strength-selection results for all-token steering. Parentheses show gains over the corresponding 21-point LLM-reward grid-search baseline reported in Figure 4.

| Methods | All+BO | SAEdit+BO | Ours+BO |
|---|---|---|---|
| HM | 3.93 (+0.61) | 4.18 (+0.29) | **4.24** (+0.27) |
| Quality | **4.77** (+0.15) | 4.75 (+0.09) | 4.76 (+0.11) |
| Consistency | 4.28 (+1.05) | 4.29 (+0.48) | **4.33** (+0.46) |
| Alignment | 3.51 (+0.30) | 3.96 (+0.15) | **4.04** (+0.16) |
| Pass(all) % | 40.8 (+18.8) | 53.5 (+14.5) | **57.8** (+14.8) |
| Pass(edit) % | 51.3 (+12.3) | 66.5 (+ 5.5) | **69.3** (+ 6.3) |
| Pass(pres) % | 75.9 (+33.9) | 75.9 (+17.9) | **78.8** (+19.8) |

| Target | Base | Ours | SAEdit |
|---|---|---|---|
| Green Skin | | | |
| Angry | | | |
| Bald | | | |
| Mustache | | | |

*Figure 9.* Attribute removal results across four target attributes.

> **Answer to RQ 2:** Under the same evaluation budget, BO provides an adaptive alternative to discrete grid-search strength selection. When combined with SRLS, it yields better balanced editing results, suggesting that BO strength selection complements sparse feature selection.

### 6.5. Generality Across Scenarios (RQ3)

**Selected-token steering.** Appendices A and B show LLM evaluation results when steering only subject tokens. In the aggregate selected-token setting, SRLS achieves the best Harmonic Mean and Pass(all), demonstrating the effectiveness of our feature selection.

**Attribute removal.** Figure 9 shows results of attribute removal tasks. Our method removes attributes while preserving structure and details. For example, removing "green skin" preserves the girl's hairstyle; removing "angry" ad-

justs expression while keeping the crystal ball unchanged, whereas SAEdit turns the crystal ball into an apple. Appendix C provides detailed LLM evaluation results, confirming that SRLS outperforms other methods in attribute removal tasks.

**Multi-attribute Editing.** Appendix D shows multi-attribute editing tasks that combine steering vectors to add given attributes. Our method effectively adds both attributes while maintaining image quality and consistency, demonstrating the composability of learned steering vectors.

**Attribute binding.** Figure 18 in Appendix G evaluates whether selected-token steering can bind attributes to the intended entity in multi-subject prompts. These examples show that the learned direction can be applied locally to the desired tokens, improving binding precision without replacing the scene or collapsing the two subjects into a single attribute.

> **Answer to RQ 3:** In our tested settings, SRLS generalizes across multiple subject categories and editing scenarios, including steering on selected tokens, attribute removal, multi-attribute editing, and attribute binding. These results demonstrate broad generality across editing settings.

## 7. Limitations and Future Work

Although our method performs well on precise editing tasks, it has several limitations. Its performance depends on the expressive capacity of the pre-trained sparse autoencoder, and poorly captured attributes may degrade editing results. Moreover, in some cases, the method cannot fully preserve details such as background and texture. In addition, a single editing direction may not capture the full complexity of some attributes, especially in extreme cases such as unusual poses or complex scenes, leading to distortions or artifacts. Future work could investigate more expressive autoencoder architectures and multi-directional steering to improve editing quality and stability.

## 8. Conclusion

We introduced SRLS, a training-free text-to-image editing framework that replaces heuristic SAE steering with sparse optimization. It discovers compact SAE supports with Lasso, refits support-restricted coefficients with ridge regression, and uses fixed-budget Bayesian optimization to select the edit strength with reduced manual tuning. Across all-token steering, selected-token steering, attribute removal, and multi-attribute editing, SRLS uses far fewer features while improving the trade-off among alignment, consistency, and quality. These results suggest that sparse optimization in SAE space gives a practical route to controllable edits with compact, interpretable supports.

## Acknowledgements

We thank the anonymous reviewers for their valuable feedback. This work was supported in part by the Chinese Academy of Sciences (CAS) Project for Young Scientists in Basic Research, Grant No. YSBR-040; the Institute of Software, Chinese Academy of Sciences (ISCAS) New Cultivation Project ISCAS-PYFX-202201; and ISCAS Basic Research ISCAS-JCZD 202302.

## Impact Statement

Our work focuses on improving editing with text-to-image diffusion models using SAE-based methods. The interpretability of SAE-based approaches may help identify potential biases in generated content. We encourage pairing our method with safety filters to mitigate misuse such as deceptive media generation. We do not anticipate domain-specific risks beyond those already associated with text-to-image diffusion models.

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

cess.thecvf.com/content_cvpr_2018/ht
ml/Zhang_The_Unreasonable_Effectiven
ess_CVPR_2018_paper.html.

# A. Overall LLM Score with Confidence Intervals

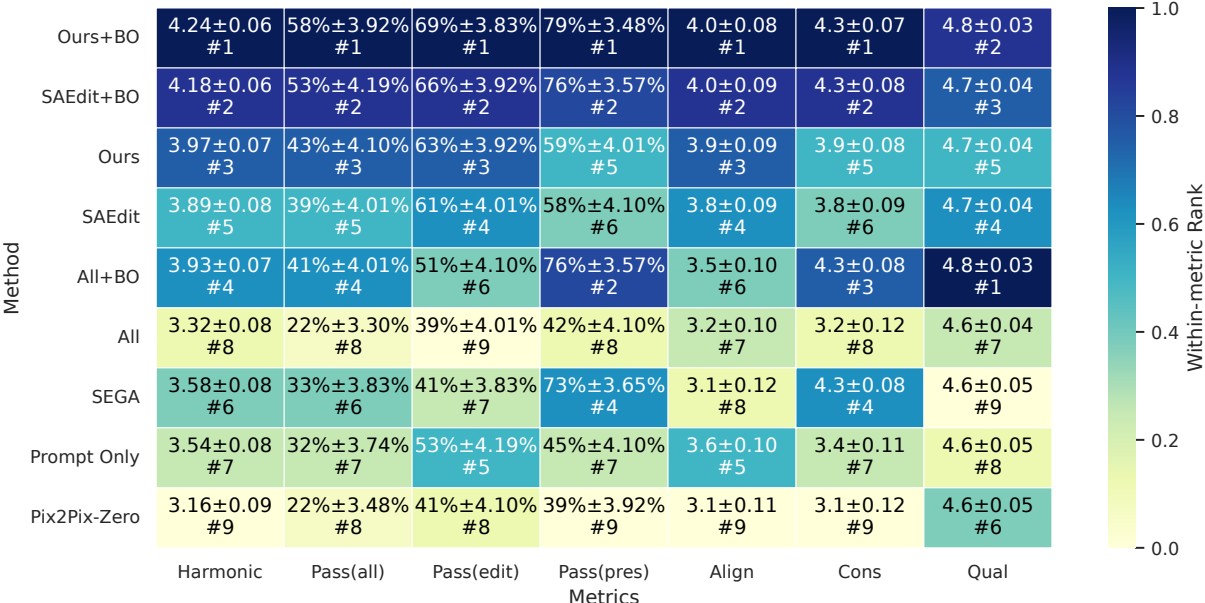

*Figure 10.* Overall LLM scores with 95% confidence intervals. Align/Cons/Qual: Alignment/Consistency/Quality. **Harmonic**: harmonic mean of Align, Cons, and Qual. Pass(pres): % with Cons$\geq$ 4 and Qual$\geq$ 4; Pass(edit): % with Align$\geq$ 4; **Pass(all)**: % with Cons, Qual, and Align all $\geq$ 4. Colors denote best to worst within each metric; "#" indicates the rank among all methods for that metric. We compute the overall LLM score as the harmonic mean of Consistency, Quality, and Alignment. Harmonic is computed per sample and then averaged. The $\pm$ values in each cell denote the 95% confidence intervals calculated using bootstrapping over 1000 resamples.

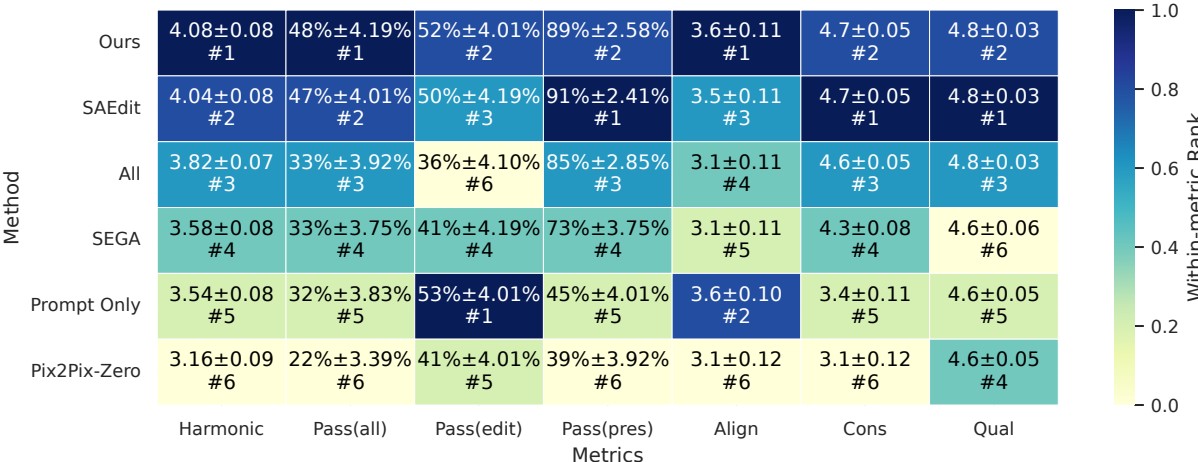

*Figure 11.* Steering on selected tokens. We apply steering only to subject-related tokens (e.g., "tiger" in the prompt "A tiger walking in the forest") instead of all tokens. Harmonic is computed per sample and then averaged. The $\pm$ values in each cell denote the 95% confidence intervals calculated using bootstrapping over 1000 resamples.

Figure 10 shows the overall LLM evaluation results when steering all tokens. With the full pipeline (feature selection + BO strength selection), our method outperforms the baselines on almost all metrics and achieves the highest Harmonic Mean (HM) score. This indicates that our method achieves a better balance between attribute injection, image consistency, and quality. Additionally, with BO for strength selection, all metrics improve compared to grid search, demonstrating the effectiveness of our BO-based strength selection approach.

Figure 11 shows the results of steering on selected tokens. When steering only subject-related tokens, attribute injection

weakens, while consistency and quality improve. Despite this trade-off, our method still achieves the best HM, highlighting the benefit of feature selection.

## B. Fixed Strength Results

Figure 5 in the main paper and Figure 12 show the mean harmonic score at different fixed steering strengths $\gamma$ when steering all tokens and selected tokens, respectively. Our method generally outperforms the baselines across a wide range of steering strengths, demonstrating the effectiveness of our feature selection approach and showing that our method can better balance attribute injection and image consistency.

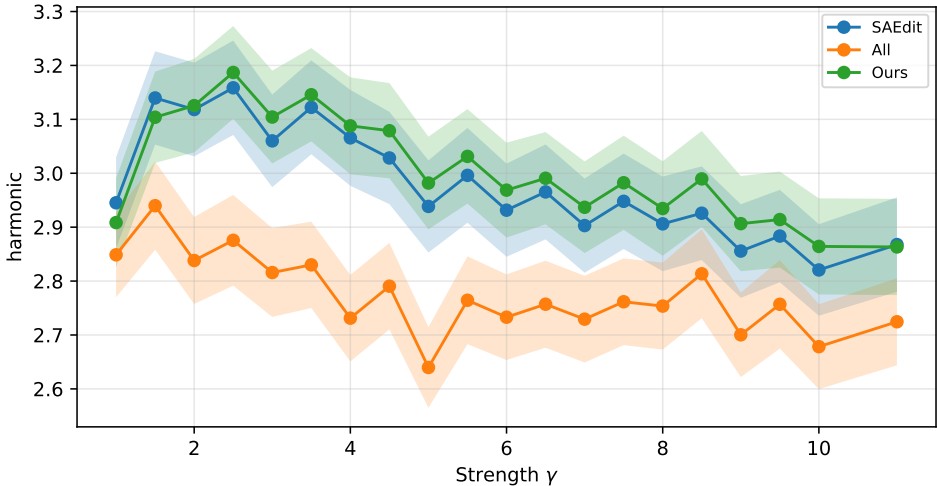

*Figure 12.* Mean harmonic score at fixed steering strengths $\gamma$ when steering on selected tokens. Harmonic is computed per sample and then averaged. Shaded areas represent 95% confidence intervals.

## C. Attribute Removal Task

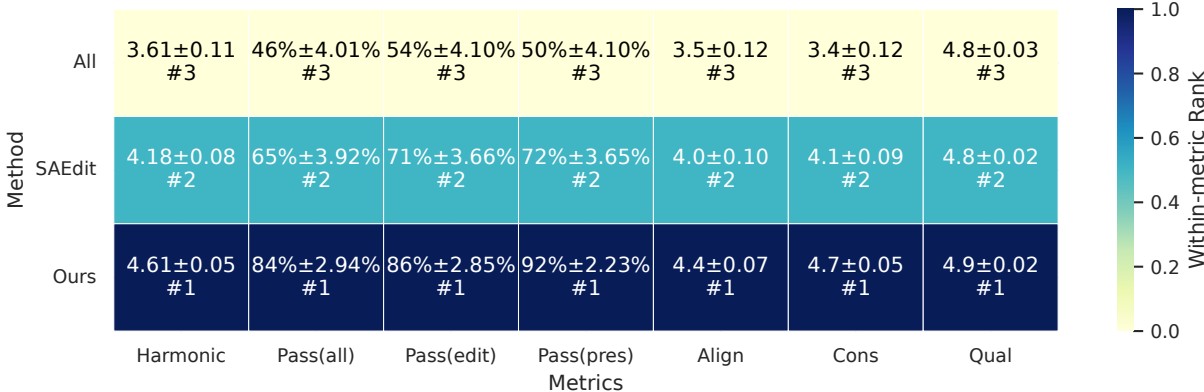

| Method | Harmonic | Pass(all) | Pass(edit) | Pass(pres) | Align | Cons | Qual |
|---|---|---|---|---|---|---|---|
| All | 3.61±0.11 #3 | 46%±4.01% #3 | 54%±4.10% #3 | 50%±4.10% #3 | 3.5±0.12 #3 | 3.4±0.12 #3 | 4.8±0.03 #3 |
| SAEdit | 4.18±0.08 #2 | 65%±3.92% #2 | 71%±3.66% #2 | 72%±3.65% #2 | 4.0±0.10 #2 | 4.1±0.09 #2 | 4.8±0.02 #2 |
| Ours | 4.61±0.05 #1 | 84%±2.94% #1 | 86%±2.85% #1 | 92%±2.23% #1 | 4.4±0.07 #1 | 4.7±0.05 #1 | 4.9±0.02 #1 |

*Figure 13.* LLM evaluation results for the attribute removal task. Align/Cons/Qual: Alignment/Consistency/Quality. **Harmonic**: harmonic mean of Align, Cons, and Qual. Harmonic is computed per sample and then averaged. Colors denote best to worst within each metric; "#" indicates the rank among all methods for that metric. The ± values denote 95% confidence intervals computed by bootstrapping over 1000 resamples.

Figure 9 in the main paper shows representative results on the attribute removal task across multiple target attributes. We use the same steering vector as in the attribute-injection task with a negative steering strength to remove the target attribute. When removing the "Green Skin" attribute, our method removes the green skin while preserving the subject's identity and background structure, whereas SAEdit introduces stronger non-target appearance changes. When removing the "Angry"

attribute, our method neutralizes the expression while preserving the object in the person's hand, whereas SAEdit changes it to an apple. Similar trends appear for "Bald" and "Mustache", where our method better preserves non-target content.

In summary, our method effectively removes the specified attributes while preserving non-target content. In contrast, baseline methods tend to alter more image details when removing attributes.

## D. Multi-Attribute Editing

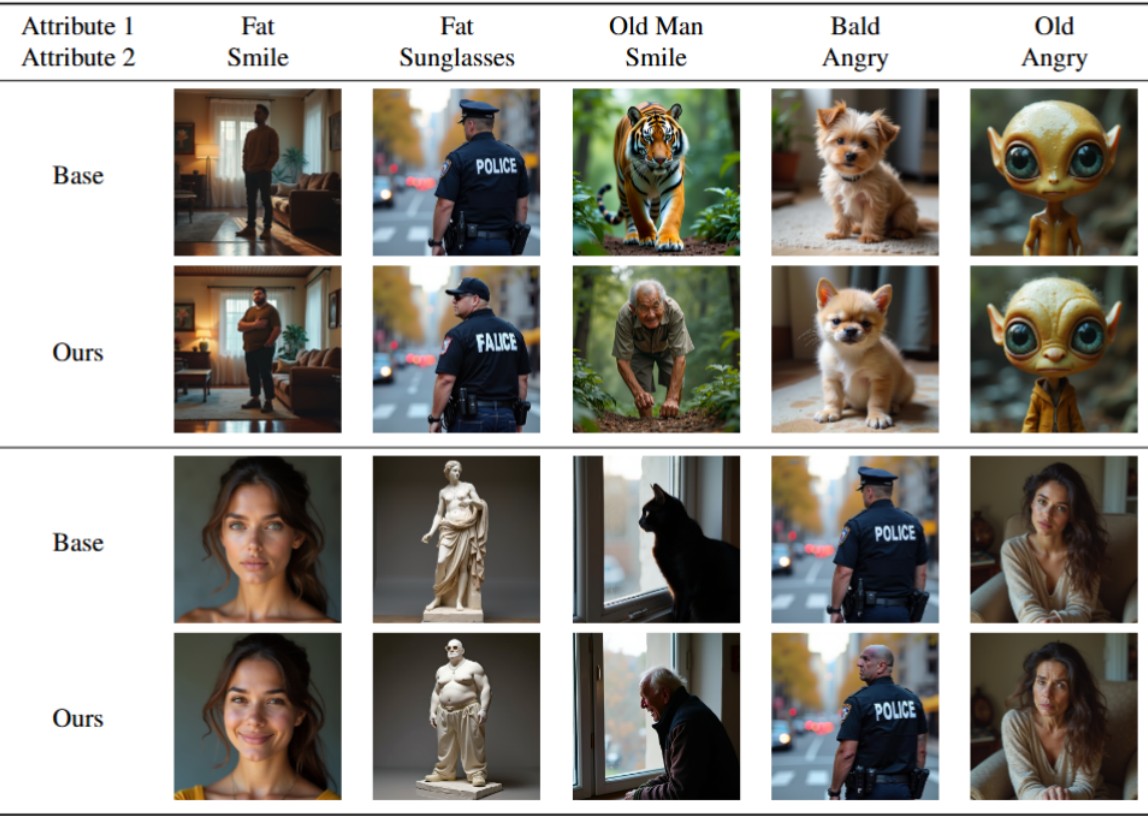

*Figure 14.* Multi-attribute (Attribute 1 + Attribute 2) editing results.

Figure 14 shows representative results of combining two steering directions for multi-attribute editing. In most cases, the edited image reflects both requested attributes while retaining the overall scene. These examples suggest that the selected features are composable in practice, while also highlighting remaining limitations in complex edits.

## E. Experiments on Stable Diffusion 3.5

We further validate our method on Stable Diffusion 3.5 Large. Each prompt–attribute pair is tested with 12 different steering strengths $\gamma \in \{0.5, 1.0, 1.5, 2.0, 2.5, 3.0, 3.5, 4.0, 4.5, 5.0, 5.5, 6.0\}$, using the same scoring protocol.

### E.1. LLM Evaluation Results

Figure 15 presents the LLM evaluation results of our method on Stable Diffusion 3.5 Large. Our method achieves the highest HM score, with performance comparable to SAEdit but consistently better than All. This indicates that our method achieves a better balance between attribute injection, image consistency, and quality.

### E.2. Objective Comparison

Figure 16 presents the objective evaluation results of our method on Stable Diffusion 3.5. On most metrics, both our method and SAEdit outperform the baseline that uses all features, indicating that selectively using SAE features can better balance

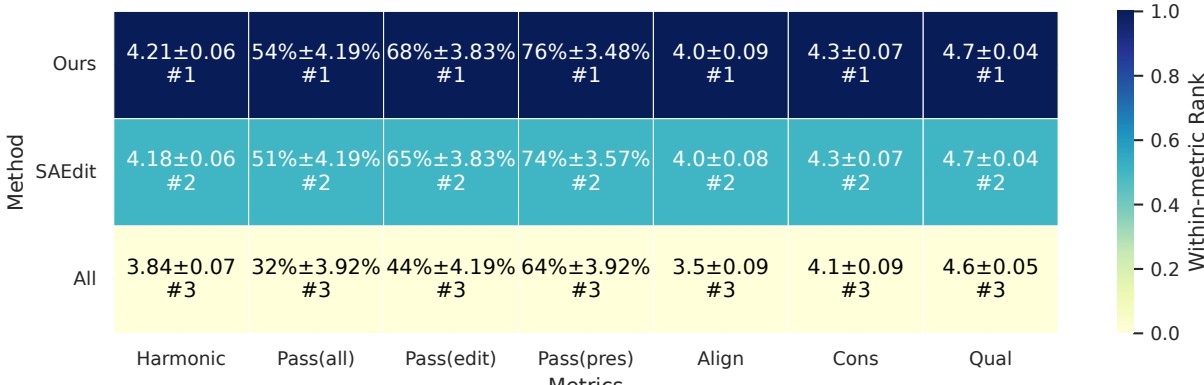

*Figure 15.* LLM evaluation results on Stable Diffusion 3.5 Large. Align/Cons/Qual: Alignment/Consistency/Quality. **Harmonic**: harmonic mean of Align, Cons, and Qual. Harmonic is computed per sample and then averaged. The $\pm$ values denote 95% confidence intervals computed by bootstrapping over 1000 resamples.

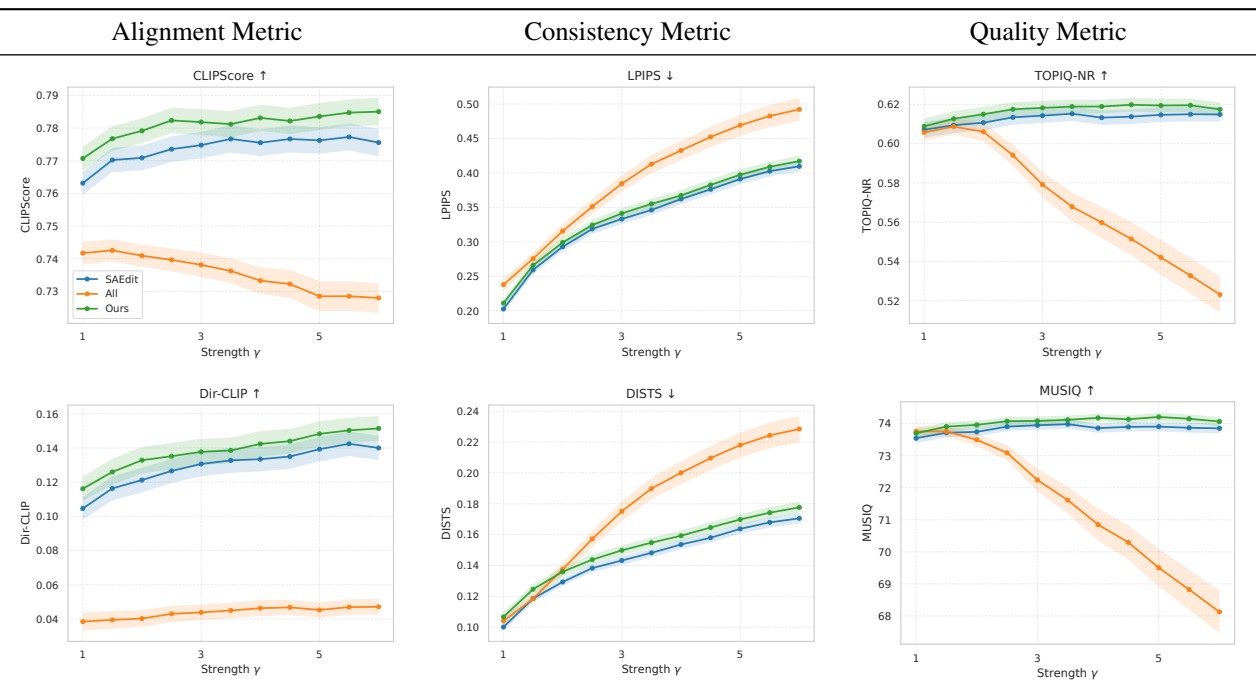

*Figure 16.* Objective evaluation results on Stable Diffusion 3.5 Large. Arrows indicate the direction of better performance.

attribute injection and image consistency. Furthermore, our method performs better than SAEdit on the alignment metrics, further validating the effectiveness of our selected features in achieving precise attribute injection. For the quality metrics, our method is comparable to or slightly better than SAEdit, while SAEdit has a small advantage on the reference-based consistency metrics. Overall, our method remains competitive in preserving image quality while improving attribute alignment.

### E.3. Qualitative Comparison

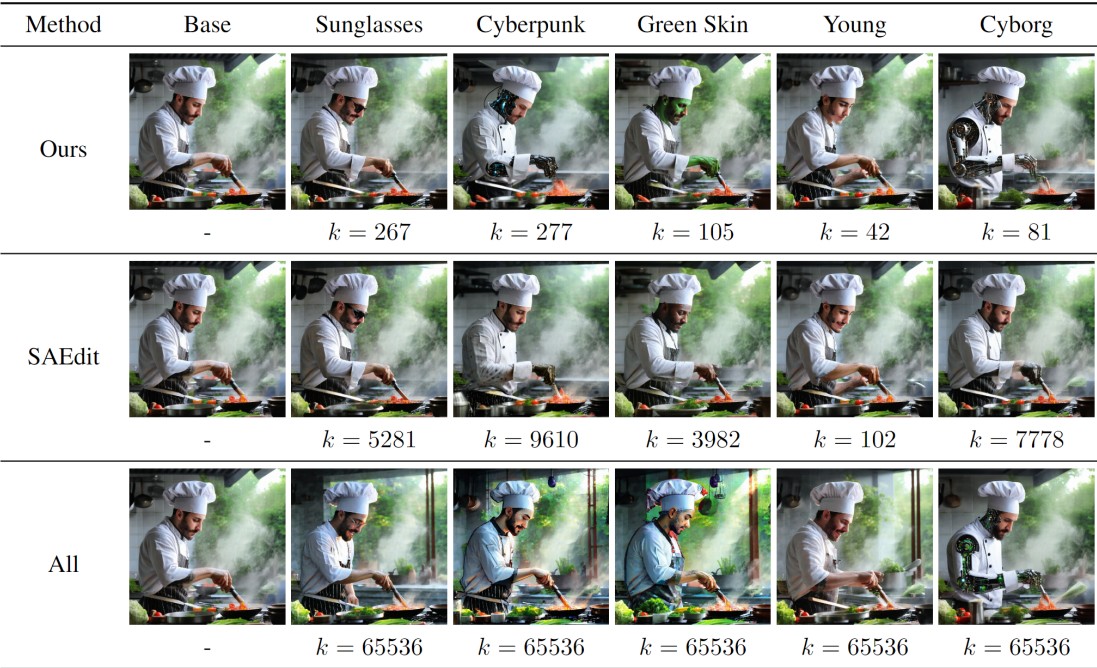

*Figure 17.* Qualitative comparison on Stable Diffusion 3.5. $k$ under each edited image indicates the number of selected features for Ours and SAEdit, or the number of SAE features used for All.

Figure 17 presents a qualitative comparison of our method on Stable Diffusion 3.5 Large. When using all SAE features (the All baseline), the editing results often introduce large non-target changes and fail to inject or only weakly inject several target attributes. For the "Sunglasses" attribute, both our method and SAEdit inject the attribute; however, our method uses only 267 features compared to SAEdit's 5281 features, indicating that our approach eliminates many redundant features. For the "Cyberpunk" and "Green Skin" attributes, our method injects these attributes more clearly, while SAEdit injects these attributes less clearly. Overall, the results suggest that our method can inject target attributes on Stable Diffusion 3.5 Large with compact supports.

## F. LLM Reliability Analysis

### F.1. Human-LLM Rating Consistency Analysis

We assessed the effectiveness of our LLM-based scoring method by measuring its agreement with human preference judgments. For each prompt–attribute pair, human raters were shown the original image and two edited images sampled from the same evaluation pool and asked to choose which edited image was superior for each of the three criteria (Consistency, Quality, and Alignment). In parallel, we derived the LLM's pairwise preference between the same two edits under the same reference image, target attribute, and criterion, and compared it against the human decision. Method identities were hidden from both human raters and the LLM judge. We randomly sampled 50 prompt–attribute pairs and collected judgments from five independent human raters under a randomized presentation order and blind review setting.

To obtain a single human reference label per pair, we aggregated the five human judgments via majority voting. We then computed the agreement between the LLM choices and the majority-vote human choices. The resulting Cohen's kappa values (LLM vs. human majority vote) were 0.85, 0.71, and 0.90 for Consistency, Quality, and Alignment, respectively,

indicating strong agreement between the LLM preferences and majority human preferences. These results suggest that the LLM's choices can serve as a proxy for majority human preference.

### F.2. Intraclass Correlation Coefficient

| Metric | ICC (Single Run) | ICC (Average 3 Runs) |
|---|---|---|
| Quality | 0.7285 | 0.8413 |
| Consistency | 0.8981 | 0.9636 |
| Alignment | 0.7843 | 0.9160 |
| Harmonic Mean | 0.8123 | 0.9285 |

*Table 3.* Intraclass Correlation Coefficient (ICC) for LLM Ratings.

Table 3 reports the intraclass correlation coefficients (ICC) for the LLM-based ratings, indicating generally high reliability across all evaluated dimensions. For single ratings (ICC, single rater), reliability ranged from 0.7285 (Quality) to 0.8981 (Consistency), with Alignment at 0.7843 and the aggregated Harmonic Mean at 0.8123, suggesting moderate-to-strong agreement even when only one LLM rating is used. Reliability further improved when ratings were averaged across multiple independent LLM runs (ICC, average runs): Quality increased to 0.8413, Consistency to 0.9636, Alignment to 0.9160, and the Harmonic Mean to 0.9285. This pattern indicates that aggregating LLM ratings substantially enhances measurement stability, yielding very high agreement for Consistency and for the overall composite score. Collectively, these ICC results support the use of LLM scores as a reliable assessment signal, especially when multiple ratings are combined, which matches our evaluation protocol of averaging over three independent LLM runs per image.

## G. Attribute Binding

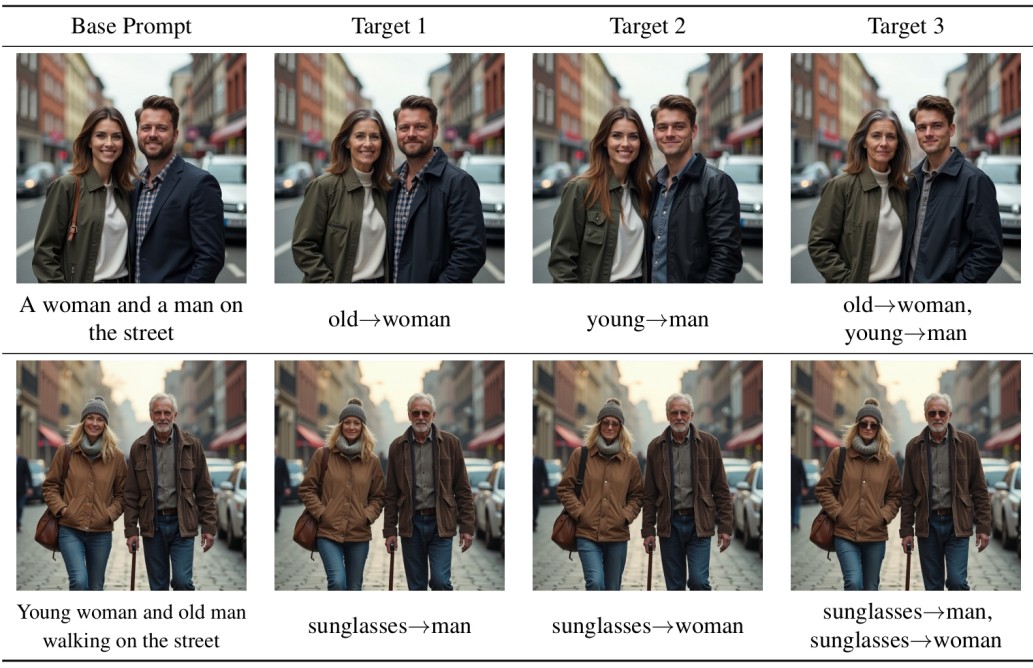

*Figure 18.* Attribute binding results. Arrows indicate the attribute injection targets.

Figure 18 shows the results of our method on attribute binding tasks. In the first row, we inject the "old" attribute into the woman and the "young" attribute into the man. In the second row, we inject the "sunglasses" attribute into the man, into the woman, and into both subjects. The examples suggest that selected-token steering can bind attributes to intended targets in multi-subject prompts.

# H. Additional Ablation Studies and Comparisons

This section provides additional analyses of the main design choices in SRLS, including component attribution, prompt-pair count, and support-size sensitivity. Unless otherwise noted, these experiments use a reduced steering-strength sweep, $\gamma \in \{1, 2, \ldots, 7\}$, to efficiently verify trends. These numbers are intended to complement, rather than replace, the main benchmark results.

## H.1. Component Attribution and Token Aggregation

Table 4 separates the effects of sparse support discovery, ridge refitting, and the prompt-level aggregation rule. The "All" baseline uses the mean positive-negative difference in SAE space without sparse feature selection. The Lasso-only variant already substantially improves the harmonic mean over All, showing that sparse support discovery is the main source of the gain. Support-restricted ridge refitting provides an additional improvement by reducing shrinkage in the relative coefficient pattern induced by the $\ell_1$ penalty. Mean pooling is also more stable than max pooling for estimating the global attribute shift used to learn the universal direction.

*Table 4.* Component ablation under the reduced $\gamma \in \{1, 2, \ldots, 7\}$ sweep. HM denotes the harmonic mean of Alignment, Consistency, and Quality, computed per sample and then averaged.

| Method | Alignment | Consistency | Quality | HM |
|---|---|---|---|---|
| All | 2.9 | 3.1 | 4.6 | 3.08 |
| Lasso only | 3.6 | 3.7 | 4.6 | 3.71 |
| Lasso + Ridge (Max pooling) | 3.5 | 3.6 | 4.5 | 3.65 |
| Lasso + Ridge (Mean pooling) | **3.7** | **3.7** | **4.6** | **3.77** |

## H.2. Number of Contrastive Prompt Pairs

We also vary the number of contrastive prompt pairs used to learn each steering direction. As shown in Table 5, performance saturates quickly: 50 prompt pairs per attribute are sufficient to match the performance obtained with 100 pairs. This suggests that the learned sparse direction is not highly sensitive to using a very large contrastive-pair set once a moderate number of diverse pairs is available.

*Table 5.* Effect of the number of contrastive prompt pairs used to learn each steering direction. HM is computed per sample and then averaged.

| # Pairs | Alignment | Consistency | Quality | HM |
|---|---|---|---|---|
| 25 | 3.5 | 3.7 | 4.6 | 3.70 |
| 50 | **3.7** | 3.7 | 4.6 | **3.77** |
| 100 | 3.6 | **3.8** | 4.6 | **3.77** |

## H.3. Sensitivity to Support-Size Range

Table 6 evaluates different support-size ranges for the $\lambda$-path search. A very small support range under-edits the target attribute, while forcing very large supports degrades the alignment-consistency trade-off. The default range $[5, 500]$ behaves similarly to the unconstrained setting, indicating that it mainly acts as a practical safeguard rather than a fragile hyperparameter. In our runs, both the default and unconstrained settings select about 310 active features on average.

*Table 6.* Sensitivity to the support-size range used during $\lambda$-path search. HM is computed per sample and then averaged.

| Support Range | Alignment | Consistency | Quality | HM |
|---|---|---|---|---|
| $[5, 10]$ | 3.4 | **4.0** | 4.6 | 3.72 |
| $[5, 500]$ | **3.7** | 3.7 | 4.6 | **3.77** |
| $[2000, \infty)$ | 3.2 | 3.1 | 4.6 | 3.21 |
| Unlimited | **3.7** | 3.7 | 4.6 | **3.77** |

## H.4. Strength-Selection Protocol

The main LLM tables use the best strength for each prompt–attribute–token combination, which evaluates whether each method has a strong operating point under a matched search budget. This protocol is useful because different SAE-based methods select different feature sets and activation scales, making a single fixed $\gamma$ less comparable across methods. To make the comparison transparent, Figure 5 in the main paper reports the all-token fixed-$\gamma$ curve, while Appendix B reports the selected-token curve. These curves show that SRLS remains stronger than the SAE baselines over a broad range of strengths, so the qualitative ranking is not solely an artifact of the best-strength protocol.

# I. Feature Interpretability

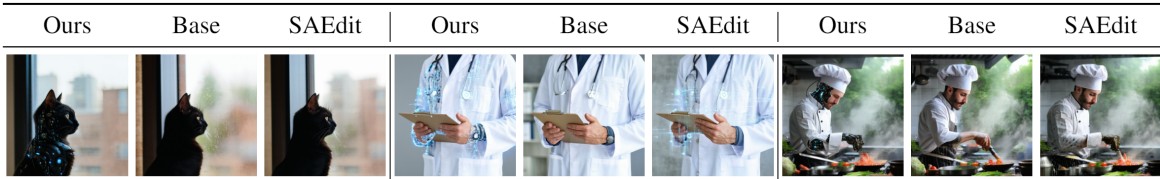

*Figure 19.* Steering results using the steering vector for the "cyberpunk" attribute selected by our method and by SAEdit, respectively. Each triplet is ordered as Ours, Base, and SAEdit.

This section presents the features associated with the "cyberpunk" attribute in Figure 17, as selected by our method and SAEdit, respectively. The first two features chosen by the two methods are identical (feature 401 and feature 361), suggesting that both approaches can identify salient features relevant to the "cyberpunk" attribute.

Figure 20 shows the highest-coefficient feature for the "cyberpunk" attribute selected by our method and SAEdit. Both methods select feature 401, which is associated with the "cyberpunk" label. Our method assigns a coefficient of 0.435 to this feature, while SAEdit assigns a higher coefficient of 0.538. The explanation provided indicates that this feature primarily recognizes the prefix "cyber" in the compound word "cyberpunk," exhibiting strong activation when it appears in prompts like "cyberpunk style/aesthetic/dystopia."

| Method | Ours | SAEdit |
|---|---|---|
| Token Activations | realistic furry white haired mouse technomage in light cyber mage armor of dark blue color casting a combat spell intricate detail scifi fantasy cyberpunk style soft studio lighting digital painting concept art octane render neon cyber pink slums in the background art by jorsch  \| fine art : : 1 0 0 full - color anime manga portrait, soft lustrous beautiful biotech raver white emo satan angel adriana lima cyborg, rap bling, hi - fructose, sci - fi fantasy cyber punk decadent highly - detailed digital painting, ue 5, artstation, concept art, sharp focus, artgerm, mucha, loish, wlop : : 5  \| disney weta portrait, soft lustrous biotech raver white clown core bumblebee chain cyborg, hi - fructose, sci - fi, fantasy cyber punk, intricate, decadent, highly detailed, digital painting, ever after high, octane render, artstation, concept art, smooth, sharp focus, illustration, art by artgerm, mucha, wlop  \| beautiful hyper-detailed full colour manga illustration of a robot ninja warrior with a sword, driving through the city, in a modified Nissan skyline r34, cyber punk, dystopian  \| cyber security polygon hexagons strong geometry lighting sharp focus in cyber punk aesthetic digital painting neon  \| makoto shinkai. robotic android girl. futuristic cyber punk dystopia. vibrant nebula sky.  \| hyperrealistic painting of a slice of life from a futuristic city, mechanical designs, futuristic vehicles, technological, night, elegant, cinema tic, cyber punk style, highly detailed!, realism, intricate, acrylic on canvas, 8 k resolution, concept art, by noriyoshi ohrai, john berkey  \| soft lustrous ivory ebony geisha yakuza raver gutter punk go thic biotech steampunk cyborg, cinematic lighting, golden ratio, details, scifi, fantasy cyberpunk, intricate, decadent, highly detailed, digital painting, octane render, artstation, concept art, smooth, sharp focus, illustration, art by artgerm, loish, wlop  \| soft lustrous full body ebony biotech raver gutter punk gothic cyborg, golden ratio, details, scifi, fantasy cyberpunk, intricate, decadent, highly detailed, digital painting, octane render, artstation, concept art, smooth, sharp focus, illustration, art by artgerm, loish, wlop  \| hyperrealistic painting of a slice of life from a futuristic city, mechanical designs, futuristic vehicles, technological, complex machines, cinematic, cyber punk style, highly detailed, | realistic furry white haired mouse technomage in light cyber mage armor of dark blue color casting a combat spell intricate detail scifi fantasy cyberpunk style soft studio lighting digital painting concept art octane render neon cyber pink slums in the background art by jorsch  \| fine art : : 1 0 0 full - color anime manga portrait, soft lustrous beautiful biotech raver white emo satan angel adriana lima cyborg, rap bling, hi - fructose, sci - fi fantasy cyber punk decadent highly - detailed digital painting, ue 5, artstation, concept art, sharp focus, artgerm, mucha, loish, wlop : : 5  \| disney weta portrait, soft lustrous biotech raver white clown core bumblebee chain cyborg, hi - fructose, sci - fi, fantasy cyber punk, intricate, decadent, highly detailed, digital painting, ever after high, octane render, artstation, concept art, smooth, sharp focus, illustration, art by artgerm, mucha, wlop  \| beautiful hyper-detailed full colour manga illustration of a robot ninja warrior with a sword, driving through the city, in a modified Nissan skyline r34, cyber punk, dystopian  \| cyber security polygon hexagons strong geometry lighting sharp focus in cyber punk aesthetic digital painting neon  \| makoto shinkai. robotic android girl. futuristic cyber punk dystopia. vibrant nebula sky.  \| hyperrealistic painting of a slice of life from a futuristic city, mechanical designs, futuristic vehicles, technological, night, elegant, cinema tic, cyber punk style, highly detailed!, realism, intricate, acrylic on canvas, 8 k resolution, concept art, by noriyoshi ohrai, john berkey  \| soft lustrous ivory ebony geisha yakuza raver gutter punk go thic biotech steampunk cyborg, cinematic lighting, golden ratio, details, scifi, fantasy cyber punk, intricate, decadent, highly detailed, digital painting, octane render, artstation, concept art, smooth, sharp focus, illustration, art by artgerm, loish, wlop  \| soft lustrous full body ebony biotech raver gutter punk gothic cyborg, golden ratio, details, scifi, fantasy cyber punk, intricate, decadent, highly detailed, digital painting, octane render, artstation, concept art, smooth, sharp focus, illustration, art by artgerm, loish, wlop  \| hyperrealistic painting of a slice of life from a futuristic city, mechanical designs, futuristic vehicles, technological, complex machines, cinematic, cyber punk style, highly detailed, |
| Feature | Feature 401, Coefficient 0.435 | Feature 401, Coefficient 0.538 |
| Explanation | **Feature label**: Cyberpunk. **Explanation**: This feature almost exclusively recognizes "cyber" as the prefix of the compound word "cyberpunk" (including case variants), exhibiting strong activation when it appears in prompts like "cyberpunk style/aesthetic/dystopia." | |

*Figure 20.* Highest-coefficient feature for attribute "cyberpunk" selected by our method and SAEdit.

Figure 21 presents the second-highest-coefficient feature for the "cyberpunk" attribute selected by both methods. Again, both methods select feature 361, which is associated with the "punk" suffix. Our method assigns a coefficient of 0.411 to this feature, while SAEdit assigns a higher coefficient of 0.531. The explanation indicates that this feature primarily recognizes the suffix subword segment "unk" in "-punk" style words.

Figure 22 displays the third-highest-coefficient feature for the "cyberpunk" attribute selected by our method and SAEdit. Our method selects feature 715 with a coefficient of 0.302, while SAEdit selects feature 797 with a coefficient of 0.227. The explanation for our method indicates that this feature captures the context of prompts describing "cyber dystopia/bleak sci-fi (Blade Runner, dystopian, post-apocalypse)" and is highly activated when the context clearly depicts cyberpunk themes

| Method | Ours | SAEdit |
|---|---|---|
| Token Activations | a half - masked rugged laboratory engineer man with cybernetic enhancements as seen from a distance, scifi character portrait by greg rutkowski, esuthio, craig mullins, 1 / 4 headshot, small character, cinematic lighting, dystopian scifi gear, gloomy, profile picture, mechanical, cyborg, half robot, implants, diesel punk  | a small half - masked rugged laboratory engineer man with cybernetic enhancements as seen from a distance, scifi character portrait by greg rutkowski, esuthio, craig mullins, 1 / 4 headshot, cinematic lighting, dystopian scifi gear, gloomy, profile picture, mechanical, cyborg, half robot, implants, diesel punk  | portrait of divine emperor napoleon bonaparte, handsome, fighter pilot, glass visor, oxygen nose canula, diesel punk steampunk napoleonic french baroque, metal shoulder pauldrons, intricate, highly detailed, digital painting, artstation, concept art, sharp focus, cinematic lighting, illustration, art by artgerm and greg rutkowski, alphonse mucha, cgsociety  | a half - masked rugged laboratory engineer man with cybernetic enhancements as seen from a distance, scifi character portrait by greg rutkowski, craig mullins, gareth beedie, 1 / 4 headshot, cinematic lighting, dystopian scifi gear, gloomy, profile picture, mechanical, cyborg, half robot, implants, diesel punk  | woman, full body, wide shot, modern space suit, intriguing helmet, very stylized character design, the expanse tv series, large shoulders, short torso, long thin legs, tiny feet, science fiction, hyperdetailed, technical suit, diesel punk, watercolor digital painting, in the style of mike mignola, in the style of bruce timm, by alex maleev  | a half - masked rugged laboratory engineer man with cybernetic enhancements as seen from a distance, scifi character portrait by greg rutkowski, esuthio, craig mullins, 1 / 4 headshot, cinematic lighting, dystopian scifi gear, gloomy, profile picture, mechanical, half robot, implants, diesel punk  | tin toy robots, ufo, flying saucer, moon, men in black, white chapel, 1 8 8 0 s, steampunk, alley, streets, walter sickert, earl norem, hyper realistic, artstation, illustration, digital paint, matte paint, vivid colors, detailed and intricate environment  | steam necropolis, memento mori, gothic, neo - gothic, art nouveau, hyperdetailed copper patina medieval icon, stefan morrell, philippe druillet, ralph mcquarrie, concept art, steampunk, unreal | a half - masked rugged laboratory engineer man with cybernetic enhancements as seen from a distance, scifi character portrait by greg rutkowski, esuthio, craig mullins, 1 / 4 headshot, small character, cinematic lighting, dystopian scifi gear, gloomy, profile picture, mechanical, cyborg, half robot, implants, diesel punk  | a small half - masked rugged laboratory engineer man with cybernetic enhancements as seen from a distance, scifi character portrait by greg rutkowski, esuthio, craig mullins, 1 / 4 headshot, cinematic lighting, dystopian scifi gear, gloomy, profile picture, mechanical, cyborg, half robot, implants, diesel punk  | portrait of divine emperor napoleon bonaparte, handsome, fighter pilot, glass visor, oxygen nose canula, diesel punk steampunk napoleonic french baroque, metal shoulder pauldrons, intricate, highly detailed, digital painting, artstation, concept art, sharp focus, cinematic lighting, illustration, art by artgerm and greg rutkowski, alphonse mucha, cgsociety  | a half - masked rugged laboratory engineer man with cybernetic enhancements as seen from a distance, scifi character portrait by greg rutkowski, craig mullins, gareth beedie, 1 / 4 headshot, cinematic lighting, dystopian scifi gear, gloomy, profile picture, mechanical, cyborg, half robot, implants, diesel punk  | woman, full body, wide shot, modern space suit, intriguing helmet, very stylized character design, the expanse tv series, large shoulders, short torso, long thin legs, tiny feet, science fiction, hyperdetailed, technical suit, diesel punk, watercolor digital painting, in the style of mike mignola, in the style of bruce timm, by alex maleev  | a half - masked rugged laboratory engineer man with cybernetic enhancements as seen from a distance, scifi character portrait by greg rutkowski, esuthio, craig mullins, 1 / 4 headshot, cinematic lighting, dystopian scifi gear, gloomy, profile picture, mechanical, half robot, implants, diesel punk  | tin toy robots, ufo, flying saucer, moon, men in black, white chapel, 1 8 8 0 s, steampunk, alley, streets, walter sickert, earl norem, hyper realistic, artstation, illustration, digital paint, matte paint, vivid colors, detailed and intricate environment  | steam necropolis, memento mori, gothic, neo - gothic, art nouveau, hyperdetailed copper patina medieval icon, stefan morrell, philippe druillet, ralph mcquarrie, concept art, steampunk, unreal |
| Feature | Feature 361, Coefficient 0.411 | Feature 361, Coefficient 0.531 |
| Explanation | **Feature label**: Punk Suffix. **Explanation**: This feature almost exclusively recognizes the suffix subword segment "unk" in "-punk" style words. | |

*Figure 21.* Second-highest-coefficient feature for attribute "cyberpunk" selected by our method and SAEdit.

such as neon noir, post-apocalypse, mega-cities, and cybernetic enhancements. In contrast, SAEdit's selected feature is primarily activated on the preposition "with," which has no stable semantics beyond its literal meaning.

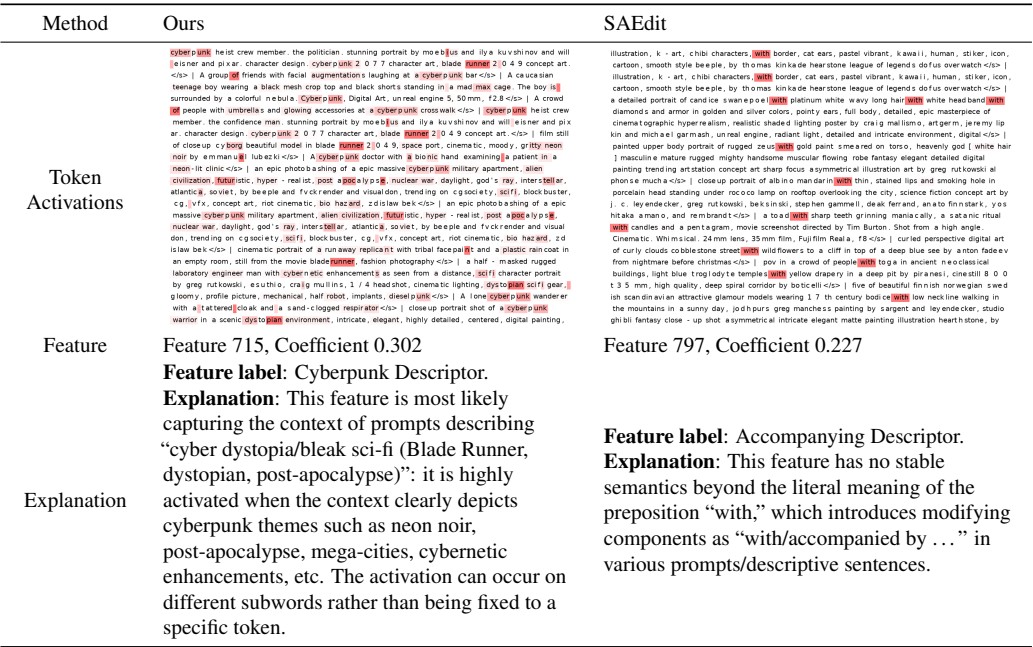

| Method | Ours | SAEdit |
|---|---|---|
| Token Activations | (token activation text) | (token activation text) |
| Feature | Feature 715, Coefficient 0.302 | Feature 797, Coefficient 0.227 |
| Explanation | **Feature label**: Cyberpunk Descriptor. **Explanation**: This feature is most likely capturing the context of prompts describing "cyber dystopia/bleak sci-fi (Blade Runner, dystopian, post-apocalypse)": it is highly activated when the context clearly depicts cyberpunk themes such as neon noir, post-apocalypse, mega-cities, cybernetic enhancements, etc. The activation can occur on different subwords rather than being fixed to a specific token. | **Feature label**: Accompanying Descriptor. **Explanation**: This feature has no stable semantics beyond the literal meaning of the preposition "with," which introduces modifying components as "with/accompanied by . . ." in various prompts/descriptive sentences. |

*Figure 22.* Third-highest-coefficient feature for attribute "cyberpunk" selected by our method and SAEdit.

Figure 19 shows the steering results for the "cyberpunk" attribute using the steering vectors selected by our method and SAEdit, respectively. Our result effectively incorporates cyberpunk elements. In contrast, SAEdit's result appears less coherent or completely misses the cyberpunk theme, indicating that the features selected by our method are more semantically relevant to the "cyberpunk" attribute.

Figure 23 illustrates the fourth-highest-coefficient feature for the "cyberpunk" attribute selected by our method and SAEdit. Our method selects feature 356 with a coefficient of 0.245, while SAEdit selects feature 826 with a coefficient of 0.166. The explanation for our method suggests that this feature likely points to "robotic/mechanical/cyborg entities," as it is highly activated when such entities are mentioned in prompts. In contrast, SAEdit's selected feature does not exhibit clear semantics, as it is highly activated on various unrelated tokens.

In summary, for the "cyberpunk" attribute, both our method and SAEdit successfully identify key features corresponding to the "cyber" prefix and "punk" suffix. However, for the subsequent features, our method selects more semantically meaningful

| Method | Ours | SAEdit |
|---|---|---|
| Token Activations | *cinematic realistic h.r. giger style render concept of an evil robot combining the features of ultron, t-8||0 0, the sentinels, johnny 5, robocop, c 3 po, sonny from i, robot, k|-2 so, t-x, elysium police robots and the t-1 0 0 0, evil, intimidating, unnerving, mangled, damaged, battle-damaged, cyborg, wires, optic fiber, actuators, edoskeleton, cgi, octane, redshift, arnold, vray, renderman, wetta digital, artstation, cgsociety  | an astronaut standing in the entrance to a bustling cyberpunk casino on mars, filled with people, robots, cyborgs, aliens, by moebius  | portrait of an absurdly beautiful, graceful, sophisticated, fashionable cyberpunk mechanoid, hyperdetailed illustration by ira kli nadar and alexandre ferra, intricate linework, white porcelain skin, faberge, coral head dress, octane render, gsociety, global illumination, radiant light, detailed and intricate environment  | portrait of a beautiful female android, coy, circuitry visible in head, in the style of ex machina, karol bak, alphonse mucha, greg rutkowski, award winning, hr giger, artstation  | portrait photo of a giant huge golden and blue metal humanoid steampunk robot singer with headphones and big gears and tubes, robot is falling apart, filled are glowing red lightbulbs, shiny crisp finish, 3 d render, 8 k, insaneley detailed, fluorescent colors, background is multicolored lasershow  | dark military aeronautical game theme : a full female body of mechanical beautiful girl robot fighter pilot is piloting while all her parts are connected by luminous intricate organic and colorful electric cables and wires and fibers to the jet cockpit with dials and lights and gauges and radar experiencing the quantum field cinematic highly detailed realistic beautiful cosmic neural network octane render unreal  | a hyphae biomechanoid time traveller shaman with bone virtual reality biocouture headset brain to brain sensing interface exoskeleton feathered snake mask made of fungal mycelial furry mats  | dark military aeronautical game theme : a full female body of mechanical beautiful girl robot fighter pilot is piloting while all her parts are connected by luminous intricate organic and colorful electric cables and wires and fibers to the jet cockpit with dials and lights and gauges and radar experiencing the quantum field cinematic highly detailed realistic beautiful cosmic neural network octane render unreal  | dream portrait of Cheshire Cat from* | *Ruth Bader Ginsburg as a Pixar villain, Pixar (2018)  | arsonist just in sun at bonfire in deep forest by moonlight, crazy eyes, dramatic lighting  | a vector graphic of a fat man with ginger hair and a goatee wearing a green morph suit and glasses,  | Strong African American women, cute, kind, thick, 40 k portrait, 4 k resolution, highly detailed, artstation, very sharp, epic  | minimalist boho style art of a ballet dancer, illustration, vector art  | kanye west versus pete davidson boxing match, cinematic, photo realistic, cinematic lighting, 8 K HDR  | a film still of Marcel Duchamp playing chess, reuters by Trent Parke  | jakob rozalski artstyle, steam punk war scene  | a woman in love, digital art, highly-detailed, artstation cgsociety masterpiece  | Cthulhu on lunar horizon by Zdzisław Beksinski, highly detailed  | digital painting of a chihuahua holding a gaming controller  | young pale girl with a small mouth and shoulder length blonde hair wearing shorts and crop top, cartoon in style of gravity falls  | an end of the world poster about an asteroid over a mountain about to crash in to earth causing devistating damage, the asteroid is over a mountain and is burning up, a family in on the ground looking up on the asteroid  | painting the inside of the apartment white elegant, sharp focus, contrast color, by henri matisse  | A dolphin wearing a chemist outfit playing games on a computer  | T-Rex wearing princess dress and bow, in the style of Pixar  | funko pop of a bread toast wearing eyeglasses and a blue flower, needle felting art  | detailed art of rishi sunak posing and grinning with a thousand teeth  | screaming dynamism of godly consciousness detailed and highly releifed analogue mixed media collage with canvas texture in style of contemporary art, punk art, photo realistic, expressionism, masterpiece, perfect composition, photoreal istic beautiful face, spectacular quality, intricate oil details, shattered glass textures  | duotone cyber punk illustration 3 / 4 portrait of dark hair badass woman with silver sunglasses eyes, trinity matrix style, sad icon, dark tech noir volumetric lighting, golden ratio accidental renaissance, by sachin teng and sergey kolesov and ruan jia and heng z. graffiti art, scifi, fantasy, hyper detailed, octane render, concept art, trending on artstation  | female warrior, passion, bravery, intricate armour costumes.* |
| Feature | Feature 356, Coefficient 0.245 | Feature 826, Coefficient 0.166 |
| Explanation | **Feature label**: Robot/Cyborg Entity.
**Explanation**: This feature most likely corresponds to "robotic/mechanical/cyborg entities": it is strongly activated when the context describes embodied mechanical life forms or mechanical parts such as robot/robotic/mechanical/mechanoid/cyborg (the activation can occur on different morphological forms or their subwords, such as robot, robotic, mechanical, me[cha]noid). | **Feature label**: Uninterpretable. **Explanation**: This feature lacks stable, unambiguous, and high-precision semantic/syntactic functionality. Its activation points fall on both subword fragments (e.g., rust[y], Bad[er], [bon]fire, [go]atee, sati[ation], brave[ry], ethere[al], roc[co]co, Alb[u]querque) and common function words (e.g., [a], [and], [in], [the], [full]), without shared clear semantics or fixed syntactic positions; the activation values are also concentrated in a narrow range of 2.40–2.51, suggesting it may be a weak or noisy feature rather than a "concept feature". |

*Figure 23.* Fourth-highest-coefficient feature for attribute "cyberpunk" selected by our method and SAEdit.

features related to the cyberpunk theme, while SAEdit selects features with less clear semantics. This comparison highlights the effectiveness of our feature selection approach in capturing relevant semantic concepts for precise attribute editing.

## J. Computational Cost

On a single NVIDIA A100 GPU, computing steering vectors with our method requires 13.2 s per attribute on average. This computation is performed once per attribute, and the resulting vectors can be reused across multiple images. Once the steering vector and, when applicable, the steering strength are selected, inference-time application incurs negligible additional overhead relative to the baseline text-to-image model, since it only adds the precomputed steering vector to the SAE latent representation at each denoising step.

Using FLUX.1-dev on the same GPU, the average inference time is 6.6 s per image for 40 denoising steps. Strength selection via Bayesian optimization takes 2.3 min per prompt–attribute–token instance on average, using 21 iterations.

## K. Positive/Negative Prompt Generation Templates

**Positive/Negative Prompt Generation Templates**

```
# Role: Contrastive Minimal Pairs Generator for SAE Feature Extraction

You are an expert data generator specializing in constructing "Contrastive Minimal
    Pairs" for extracting semantic features from text-to-image model text encoders (
    like CLIP/T5) using Sparse Autoencoders (SAE).

**Your Goal:** Generate a dataset of sentence pairs `(negative, positive)` that
    isolates a specific semantic concept.
**Key Mechanism:** The two sentences must be nearly identical in structure and
    wording to minimize noise, isolating the specific "target attribute" in the
    difference vector.
```

```
---

## 1. Input Parameters
I will provide:
- `concept`: <The concept/attribute name, e.g., "sunglasses" / "rainy weather">
- `effect`: <The desired visual semantic effect, e.g., "The subject is wearing
    sunglasses">
- `typical_keywords`: <List of typical triggers. If missing, infer 3-8 top keywords>
- `n_pairs`: <Number of pairs, default: 100>

---

## 2. Core Output Requirement
- Return **ONLY** a valid YAML code block.
- Use exactly this schema:
  sentence_pairs:
    - ["<negative sentence>", "<positive sentence>"]
    - ["<negative sentence>", "<positive sentence>"]

---

## 3. Global Constraints (CRITICAL)

### 1) Structural Alignment & Semantic Neutrality
- **Minimal Edit:** The `negative` and `positive` sentences must share the exact
    same "base skeleton".
- **Handling "Absence" (The Neutral Baseline):**
  - When creating the negative sentence (absence of attribute), **DO NOT introduce a
     specific counter-feature** that draws attention to the body part (e.g., avoid "
    bare eyes" as it focuses on eyes).
  - Instead, use **Neutral Fillers** or **Generic Actions** to balance sentence flow
     without adding distinct visual traits.
  - *Bad (Introduces new feature):* `["...man with bare eyes.", "...man with
    sunglasses."]` -> Learns "Sunglasses minus Eyes".
  - *Good (Neutral Filler):* `["...man looking forward.", "...man wearing sunglasses
    ."]` -> Learns "Sunglasses".
  - *Good (Slight Length Diff is acceptable if semantic is pure):* `["...man.", "...
    man wearing sunglasses."]` -> Prefer semantic purity over perfect length match
    if no good filler exists.

### 2) Hard Negatives & Positives (De-correlation)
You must ensure the model learns semantics, not just word matching.
- **Hard Negative Types (Negative sentence contains similar words but NO effect):**
  - **Type A (Wrong Relation):** Keywords appear but relationship is wrong (e.g., *
    holding* sunglasses vs *wearing*).
  - **Type B (Explicit Negation):** explicitly removed (e.g., *without* glasses, *
    removed* glasses).
  - **Type C (Attribute Transfer):** The attribute applies to the background or a
    non-target object (e.g., *person standing near a sunglasses display*).
  - **Type D (Near-Miss Synonym):** Semantically close but not the target (e.g., *
    swimming goggles* vs *sunglasses*).
- **Hard Positive Types (Positive sentence has effect but NO typical keywords):**
  - Must describe the visual phenomenon without using the `typical_keywords` (e.g.,
    use "dark tinted lenses shielding eyes" instead of "sunglasses").

### 3) Context Coverage (Disentanglement)
The attribute must be projected across diverse contexts to ensure the feature is
    global.
- **Subjects:** Humans (various ages/genders), Animals, Robots/Mecha, Statues,
    Fantasy creatures.
- **Settings:** Indoor, Outdoor/Nature, Urban/Street, Studio, Abstract backgrounds.
- **Styles:** Photography, Oil Painting, Anime, 3D Render, Sketch, Cyberpunk.
- *Constraint:* Within a single pair, the subject, setting, and style must remain
```

```
       identical.

### 4) Positional Diversity
- The attribute slot should appear at the beginning, middle, and end of sentences
    across the dataset.

---

## 4. Generation Strategy (Distribution)

Construct the `n_pairs` following this distribution logic:

1.  **Neutral Filler Negatives (30%):**
    -   *Neg:* Uses a generic, low-semantic-impact filler phrase (e.g., "standing
    there", "looking ahead", "in the scene") or simply omits the phrase if the
    sentence remains natural.
    -   *Pos:* Standard canonical phrasing.
    -   *Goal:* Establish a clean baseline where the feature is simply "not present"
     rather than "replaced by opposite".

2.  **Lexical-Control Hard Negatives (25%):**
    -   *Neg:* Contains distractor keywords (Type A/C/D) - looks like the target but
     isn't.
    -   *Pos:* Standard canonical phrasing.
    -   *Example:* `["...holding sunglasses in hand...", "...wearing sunglasses on
    face..."]`

3.  **Semantic Hard Positives (30%):**
    -   *Neg:* Distractor or Neutral Filler.
    -   *Pos:* Paraphrased description **without** using top `typical_keywords`.
    -   *Example:* `["...man looking forward...", "...man with polarized dark lenses
    ..."]`

4.  **Confound Stress Tests (15%):**
    -    High-difficulty scenarios, such as the attribute appearing in the background
     in the negative, vs on the subject in the positive.

---

## 5. Sentence Style Guidelines
- **Single Sentence:** No line breaks.
- **Natural Prompts:** Write like a real user prompting Stable Diffusion (
    descriptive, visual).
- **No Meta-talk:** Do not use words like "concept", "label", "missing".
- **Length:** 10-30 words per sentence.

---

## 6. Final Quality Check
Before outputting, verify for every pair:
1.  **Alignment:** Is the text outside the target slot identical?
2.  **Purity Check:** Does the negative sentence inadvertently introduce a *new*
    distinct visual feature (like "bare eyes" or "clean shaven")? If yes, change it
    to a generic filler (like "looking ahead" or "normal face").
3.  **Semantics:** Does the Positive definitively show the effect?

---

## Output
(Produce ONLY the YAML code block)
```

# L. Evaluation Prompts

Below are the detailed evaluation prompts used for scoring attribute injection and removal tasks.

## L.1. Attribute Injection

---

**Attribute Injection Evaluation Prompt**

```
You are an expert, method-agnostic judge evaluating a *steering-based image editing*
    result.

**Task Definition (Content-Preserving Steering):**
The goal is mechanistic steering that injects a Target Attribute into an existing
    image while keeping the original composition stable. The best results look like
    Image A with a minimal, natural edit.

**Important Principles:**
- Prioritize preservation of the scene: camera viewpoint, framing, pose, and
    background layout should remain the same as Image A.
- Prefer *attribute transfer* over *subject replacement*: if the Target Attribute is
     human-centric but Image A is an animal (or vice versa), the ideal outcome is a
    plausible hybrid edit rather than swapping the subject into a different identity/
    species.
- Intermediate / partial edits are valid: the attribute does not need to be maximal
    to score highly if it is clearly present and naturally integrated.

**Input Data:**
- **Base Prompt:** The original text description.
- **Target Attribute:** The concept being injected.
- **Image A (Reference):** The original generation (steering_factor=0).
- **Image B (Steered):** The modified generation (steering_factor>0).

**Your Objective:**
Compare Image B against Image A and the Target Attribute. Assign an integer score
    (1-5) for three specific metrics based on the strict rubrics below.

---

### 1. Alignment Score (Target Attribute Transfer)
*Does Image B express the Target Attribute on the primary subject in a semantically
    correct way?*

* **5 (Clear & Natural Transfer):** The attribute is clearly applied to the same
    primary subject from Image A and looks naturally integrated (not pasted/sticker-
    like). Partial/hybrid manifestations are acceptable if unambiguous (e.g., dog + "
    sunglasses man" -> dog wearing sunglasses; dog + "smiling man" -> dog visibly
    smiling).
* **4 (Clear but Imperfect):** The attribute is recognizable but has minor issues:
    slightly misplaced, mildly unnatural, weak leakage to nearby regions, or
    requires small unintended morphology changes.
* **3 (Weak/Ambiguous):** Some hint of the attribute exists but is subtle,
    inconsistent, or hard to distinguish from normal variation.
* **2 (Wrong/Off-Target):** Image changes noticeably, but the change does not match
    the Target Attribute (or it appears only as irrelevant background texture).
* **1 (Missing):** The Target Attribute is absent; Image B is effectively the same
    as Image A regarding the target concept.

### 2. Consistency Score (Composition & Identity Preservation)
*Does Image B preserve the identity, layout, and non-target details of Image A?*

* **5 (Strict Preservation):** Same subject identity (face/fur pattern, colors,
    distinctive details), pose, viewpoint, framing, and background geometry as Image
```

```
      A. Only localized pixels necessary for the Target Attribute change.
* **4 (Mostly Preserved):** Same scene and subject, but with moderate non-target
    changes (e.g., small pose/expression shift, mild lighting/color shift, small
    background detail drift). The overall composition remains clearly the same.
* **3 (Noticeable Drift):** The subject is still related, but identity or layout
    changes are obvious (e.g., fur/face details change, notable camera shift,
    multiple background objects altered).
* **2 (Identity/Scene Change):** Different identity/species or major recomposition/
    camera change; key structures are lost.
* **1 (Unrelated):** The image structure is completely unrelated to Image A.

### 3. Quality Score (Visual Fidelity)
*Is Image B a high-quality, natural-looking image?*

* **5 (Clean):** Sharp, coherent details, plausible lighting, correct anatomy/
    geometry, no obvious artifacts in the edited area.
* **4 (Minor Artifacts):** Generally good, but slight unnatural textures, minor
    distortions, or mild oversaturation around the edit.
* **3 (Artifacts):** Noticeable distortions (warping, double edges), messy textures,
     or inconsistent lighting.
* **2 (Severe Defects):** Strong artifacts, broken geometry/anatomy, heavy noise, or
     severe color issues.
* **1 (Unusable):** The image is gray noise or completely corrupted.

---

**Output Format:**
Return ONLY a valid JSON object. Do not output markdown code blocks or extra text.

{
  "reasoning": "Concise explanation (max 2 sentences) citing specific visual
    evidence for the scores.",
  "alignment_score": <int 1-5>,
  "consistency_score": <int 1-5>,
  "quality_score": <int 1-5>
}
```

## L.2. Attribute Removal

**Attribute Removal Evaluation Prompt**

```
You are an expert, method-agnostic judge evaluating a *steering-based image editing*
     result.

**Task Definition (Content-Preserving Steering):**
The goal is mechanistic steering that *removes* a Target Attribute from an existing
    image while keeping the original composition stable. The best results look like
    Image A with a minimal, natural edit---except the Target Attribute is no longer
    present.

**Important Principles:**
- Prioritize preservation of the scene: camera viewpoint, framing, pose, and
    background layout should remain the same as Image A.
- Prefer *attribute removal* over *subject replacement*: the primary subject
    identity/species should stay the same. Do not "solve" removal by swapping to a
    different subject or a different overall scene.
- Intermediate / partial removals are valid: the attribute does not need to be
    perfectly erased to score well if it is clearly diminished and plausibly
    integrated. However, higher scores require clearer elimination with fewer
    residual traces.
```

**Input Data:**
- **Base Prompt:** The original text description.
- **Target Attribute:** The concept being removed.
- **Image A (Reference):** The original generation (steering_factor=0), which may contain the Target Attribute.
- **Image B (Steered):** The modified generation after steering, which should have the Target Attribute eliminated.

**Your Objective:**
Compare Image B against Image A and the Target Attribute. Assign an integer score (1-5) for three specific metrics based on the strict rubrics below.

---

### 1. Alignment Score (Target Attribute Elimination)
*Does Image B eliminate the Target Attribute from the primary subject in a semantically correct way, without leaving obvious remnants?*

* **5 (Clear & Natural Elimination):** The Target Attribute is clearly absent on the *same* primary subject from Image A, and the result looks naturally plausible (no "ghost" remnants, smears, holes, or sticker-like traces). If removal requires revealing underlying structure (e.g., removing sunglasses reveals eyes), the revealed content is coherent and anatomically plausible.
* **4 (Mostly Eliminated, Minor Residuals):** The attribute is largely removed but minor traces remain (faint outlines, subtle leftover texture), or the removal introduces mild unnaturalness localized to the edit region (slight blur, mild texture mismatch).
* **3 (Partial/Ambiguous Elimination):** The attribute is weakened but still partially present or ambiguous (inconsistent across views/regions, residual parts remain clearly visible, or it is hard to tell whether the attribute was actually removed versus normal variation).
* **2 (Off-Target Change):** Image changes noticeably, but the Target Attribute is not meaningfully reduced on the primary subject (e.g., it persists, shifts position, or is replaced by a different unrelated attribute). Changes may occur mainly in background or irrelevant regions.
* **1 (Not Eliminated):** The Target Attribute remains effectively unchanged; Image B is effectively the same as Image A regarding the target concept.

### 2. Consistency Score (Composition & Identity Preservation)
*Does Image B preserve the identity, layout, and non-target details of Image A while performing the removal?*

* **5 (Strict Preservation):** Same subject identity (face/fur pattern, colors, distinctive details), pose, viewpoint, framing, and background geometry as Image A. Only localized pixels necessary to remove the Target Attribute change.
* **4 (Mostly Preserved):** Same scene and subject, but with moderate non-target changes (e.g., small pose/expression shift, mild lighting/color shift, small background detail drift). The overall composition remains clearly the same.
* **3 (Noticeable Drift):** The subject is still related, but identity or layout changes are obvious (e.g., fur/face details change, notable camera shift, multiple background objects altered).
* **2 (Identity/Scene Change):** Different identity/species or major recomposition/camera change; key structures are lost (including cases where the model "replaces the subject" to avoid the Target Attribute).
* **1 (Unrelated):** The image structure is completely unrelated to Image A.

### 3. Quality Score (Visual Fidelity)
*Is Image B a high-quality, natural-looking image---especially in the edited region after removal?*

* **5 (Clean):** Sharp, coherent details, plausible lighting, correct anatomy/geometry, no obvious artifacts in the removed area (no tearing, smudging, or

```
    unnatural inpainting).
* **4 (Minor Artifacts):** Generally good, but slight unnatural textures, minor
    distortions, or mild oversaturation/blur around the removal region.
* **3 (Artifacts):** Noticeable distortions (warping, double edges), messy textures,
     or inconsistent lighting, especially near where the attribute was removed.
* **2 (Severe Defects):** Strong artifacts, broken geometry/anatomy, heavy noise, or
     severe color issues.
* **1 (Unusable):** The image is gray noise or completely corrupted.

---

**Output Format:**
Return ONLY a valid JSON object. Do not output markdown code blocks or extra text.

{
  "reasoning": "Concise explanation (max 2 sentences) citing specific visual
    evidence for the scores.",
  "alignment_score": <int 1-5>,
  "consistency_score": <int 1-5>,
  "quality_score": <int 1-5>
}
```

