# OpenReview forum: "Sparse Relaxed-Lasso Steering: Automatic Sparse Autoencoder Feature Selection for Precise Image Editing"
_ICML.cc/2026/Conference — ICML 2026 regular_

### Official Review · Reviewer_aZD9 · 2026-03-09

**Soundness:** 2
**Presentation:** 2
**Significance:** 2
**Originality:** 1
**Overall Recommendation:** 3
**Confidence:** 4

**Summary:**

The paper proposed an image editing method for text-to-image diffusion models by steering sparse autoencoder (SAE) features. The approach operates on top of a pretrained SAE and a text-to-image diffusion models.

The core idea is to formulate automatic feature selection as a sparse optimization problem using Sparse Relaxed Lasso Steering (SRLS). This optimization identifies a sparse set of SAE features corresponding to the desired attribute change.

In addition, the paper proposes to automatically determine the optimal steerings strength using Bayesian optimization with a Gaussian-process surrogate model.

**Compliance With Llm Reviewing Policy:**

Affirmed.

**Final Justification:**

The rebuttal addressed most of my concerns, but I am still not fully convinced about the paper's originality and the performance of target attribute understanding in complex sentences. Overall, I will give it a score of 3.

**Key Questions For Authors:**

please see strenghts and weaknesses

**Limitations:**

yes

**Strengths And Weaknesses:**

Strengths
- The paper formulates the problem of selecting SAE features for image editing as a sparse optimization problem. This allows automatic identification of relevant features corresponding to a target attribute, improving over previous approaches that rely on manual feature selection.

- The paper is clearly wrtten and easy to follow. The optimization formulation are well presented, making the overall method accessible.

Weaknesses

- Although the paper proposed an optimization-based approach for feature selection, the overall methodology appears incremental. The method still requires optimization for each target attribute and multiple (from 50 to 150) paired prompts for each target attribute. Also, it is hard to find new insights or mechanisms beyond applying existing sparse optimization techniques to the SAE feature space.

- The proposed method introduces multiple hyperparameters (e.g., k_min, k_max, lambda_min, lambda_max, tau). It is unclear how sensitive the method is to these parameters or how well the method generalizes across different editing tasks without careful tuning.

- While the paper focuses on a precise image editing task, it is unclear how this method can ideally be compared against recent state-of-the-art editing methods (e.g., FlowAlign[1]), or evaluated on more complex editing tasks where existing methods fail.

- Since this method concentrates on the target attribute, it is not sure that the attribute and components are well binded,  such as "young woman and old man walking street" into "old woman and young man walking street" or "old woman and old man walking street", can this method successfully divide each binding?

- The optimization procedure for feature selection may introduce additional computational overhead. However, the paper does not clearly analyze the optimization cost (e.g., runtime or number of optimization steps), making it difficult to assess the practical efficiency of the approach.

[1] Kim, Jeongsol, et al. "Flowalign: Trajectory-regularized, inversion-free flow-based image editing." arXiv preprint arXiv:2505.23145 (2025).

---

> ### Author Rebuttal · Authors · 2026-03-31
>
> We thank the reviewer for the detailed feedback. We fully agree that these questions are important. To better address them, we added several small-scale ablations (using the same setting of $\gamma\in\\{1,2,\ldots,7\\}$ throughout).
>
> ### (W1) Novelty of the methodological contribution
>
> Our main contribution is the problem formulation. By using the affine SAE decoder, we turn steering vector discovery into a sparse recovery problem in the decoder space, for which Lasso is a well-established tool. In this way, we convert heuristic SAE feature selection and manual strength tuning into an interpretable, automatic, and more stable optimization pipeline in practice. We believe that a simple and effective combination is also valuable, especially in a new and practical application setting.
>
> ### (W2) Sensitivity to hyperparameters and generalization
>
> | Support Range | Alignment | Consistency | Quality | Mean Harmonic |
> | --- | --- | --- | --- | --- |
> | [5, 10] | 3.4 | 4.0 | 4.6 | 3.72 |
> | [5, 500] | 3.7 | 3.7 | 4.6 | 3.77 |
> | [2000, $\infty$] | 3.6 | 3.1 | 4.6 | 3.37 |
> | Unlimited | 3.7 | 3.7 | 4.6 | 3.77 |
>
> Our ablations show that too small a support leads to under-editing, while too large a support reduces the overall trade-off between alignment and consistency. The default support range [5, 500] mainly serves as a safeguard, rather than a hyperparameter that requires careful tuning: $k_{\min}$ and $k_{\max}$ can be set to [0, $\infty$], allowing SRLS to automatically choose a suitable support size without a clear drop in performance.
>
> In our experiments, we did not tune these parameters. We fixed $\lambda_{\min}=0.1$ and $\lambda_{\max}=2.0$, and used the same setting of $\tau$ as the baseline method SAEdit. Even under this setup, SRLS shows stable gains across multiple attributes, subjects, and models, which suggests reasonable robustness in practice.
>
> ### (W3) Comparison with state-of-the-art methods
>
> We agree that broader comparisons would strengthen the paper. During the rebuttal period, we added new experiments with FlowAlign. We used FlowAlign to edit images generated from the base prompt, provided both the base prompt and the target prompt, and used the default FlowAlign settings.
>
> The results are as follows:
>
> | Method | Alignment | Consistency | Quality | Mean Harmonic |
> | --- | --- | --- | --- | --- |
> | FlowAlign | 2.8 | 4.8 | 4.7 | 3.47 |
> | Ours+BO | 4.0 | 4.3 | 4.8 | 4.24 |
>
> The results show that FlowAlign performs well in consistency, but has a lower editing success rate, leading to a weaker overall trade-off than SRLS. We will add a discussion of FlowAlign in the revised paper and clarify the differences in task setting and evaluation protocol between FlowAlign and our method.
>
> ### (W4) Attribute-component binding
>
> This is an important issue. Our method supports selected-token steering, so in principle it can apply attribute directions to specific subject tokens. However, we have not yet systematically evaluated difficult multi-entity binding cases such as the one described by the reviewer. We will add targeted experiments in the revised paper.
>
> ### (W5) Computational efficiency
>
> **Runtime:** At inference time, SRLS only adds one tensor, so in practice its runtime is almost the same as standard inference, with nearly no extra computational cost.
>
> **Feature search and optimization time:** The feature search step is performed offline during steering vector construction and only needs to be done once. In our experiments, we used 50 steps of binary search and 200 steps of optimization, and the average feature search time was about 8.2 seconds. We believe this is acceptable as an offline cost.

---

> > ### Author Rebuttal · Reviewer_aZD9 · 2026-04-04
> >
> > Thanks to the author for their rebuttal. Most of my concerns have been resolved, except for whether the proposed method can successfully disentangle the target attribute binding. Therefore I will raise my score to 3.

---

> > > ### Author Response · Authors · 2026-04-07
> > >
> > > Thank you again for highlighting the attribute-binding issue. Following your suggestion, we conducted additional multi-entity binding experiments and uploaded representative qualitative results in an anonymous image link:
> > >
> > > https://imgur.com/a/6ukeHSO
> > >
> > > In the first row, we evaluate age-attribute binding in a two-person scene. Specifically, we inject “old” only into the female subject, “young” only into the male subject, and “old” into both subjects, respectively. In the second row, we evaluate accessory binding by injecting “sunglasses” only into the male subject, only into the female subject, and into both subjects.
> > >
> > > In these tested cases, the edited attributes are attached to the intended target subject(s) without obvious leakage to the other person. This provides additional evidence that our selected-token steering can achieve precise target-specific attribute binding in challenging multi-entity scenes. We will incorporate these additional examples and discussion into the revised version.
> > >
> > > Thank you again for raising this important point.

---

### Official Review · Reviewer_Py6S · 2026-03-11

**Soundness:** 3
**Presentation:** 3
**Significance:** 3
**Originality:** 2
**Overall Recommendation:** 4
**Confidence:** 3

**Summary:**

This paper proposes the utilization of Sparse Autoencoders (SAEs) to achieve precise, training-free steering in text-to-image (T2I) diffusion models. It casts the discovery of steering vectors as a formal sparse recovery problem. It introduces the Sparse Relaxed-Lasso Steering (SRLS) method, which employs Lasso-based optimization to identify semantically relevant features and uses Bayesian optimization (BO) to determine the optimal steering strength.

**Compliance With Llm Reviewing Policy:**

Affirmed.

**Final Justification:**

The authors addressed my concerns. I lean to weak accept.

**Key Questions For Authors:**

1. Could the authors provide the following ablation studies: (a) Lasso selection only, (b) ridge regression-based debiasing, and (c) Bayesian optimization? Relative to SAEdit, which component is the primary factor of the performance improvement?
2. Are the steering vectors discovered through the text encoder SAE still effective when the latent space is constrained by real image structures?
3. According to Section 4.4, $\lambda$ is determined via bisection search to satisfy a target sparsity range of [5, 500]. How sensitive are the final alignment and quality scores to the selection of this sparsity budget? Is there a universally optimal range across different semantic concepts, or does this range need to be manually specified for each attribute?

**Limitations:**

This paper transforms the feature selection problem into a sparse optimization problem, but it mainly applies existing mature optimization methods (Lasso and BO) systematically to the SAE control flow. Furthermore, it lacks an evaluation on real image editing.

**Strengths And Weaknesses:**

Strengths:
- S1. By leveraging the affine decoder structure of the SAE, the feature selection problem is transformed into a convex Lasso problem, which effectively resolves the inherent flaws of manual feature selection. The adoption of a ridge regression-based debiasing step to address the common shrinkage issue in standard Lasso is mathematically sound and empirically validated.
- S2. SRLS provides a plug-and-play solution without the need to fine-tune the base models.
- S3. Applying Lasso and Bayesian Optimization (BO) to the SAE structure directly achieves full automation in both feature screening and strength control.

Weaknesses:
- W1. Although the paper demonstrates in Table 2 that Bayesian Optimization (BO) outperforms grid search, it does not provide ablation studies at the core algorithmic level. The paper fails to clarify whether the performance improvement stems from the Lasso feature support set selection or the restricted ridge regression debiasing step, lacking this specific ablation study.
- W2. The current evaluation is limited to synthetic prompt editing scenarios. It lacks validation on real image editing.
- W3. The core methods of this paper (Lasso, restricted ridge regression, and Bayesian Optimization) essentially combine existing mature statistical and optimization tools. The contribution leans more towards problem formulation and engineering combination within specific application scenarios.

---

> ### Author Rebuttal · Authors · 2026-03-31
>
> We thank the reviewer for the careful reading and constructive comments. To meet the rebuttal deadline, we added some small-scale ablation studies (using the same setting of $\gamma\in\\{1,2,3,4,5,6,7\\}$ throughout) to address the questions. We respond to each point below:
>
> ### (W1 & Q1) Additional ablation studies
>
> | Method | Alignment | Consistency | Quality | Mean Harmonic |
> | --- | --- | --- | --- | --- |
> | All | 2.9 | 3.1 | 4.6 | 3.08 |
> | Lasso | 3.6 | 3.7 | 4.6 | 3.71 |
> | Lasso + Ridge | 3.7 | 3.7 | 4.6 | 3.77 |
>
> Lasso-only already brings a clear improvement over the All baseline (HM 3.71 vs. 3.08), while ridge refit provides a further gain (3.71 -> 3.77). For BO, Figure 4 and Table 2 already separate BO vs. grid search: under the same 21-evaluation budget, SRLS remains best or tied-best, and BO brings an additional gain of 0.24. Relative to SAEdit, the primary improvement appears to come from replacing heuristic feature selection with optimization-based sparse support discovery; ridge refit provides a smaller additional gain, and BO is an orthogonal practical improvement.
>
> ### (W2 & Q2) Real image editing
>
> SRLS modifies the text-conditioning representations, so in principle it can be combined with inversion- or image-conditioned editing pipelines as a plug-in conditioning control. We agree that validating this setting is important, and we will add real-image editing results in the revision.
>
> ### (W3) Novelty of the methodological contribution
>
> Our primary contribution lies in problem formulation. By leveraging the affine SAE decoder, we cast the discovery of steering vectors as a sparse recovery problem in the decoder space, with Lasso serving as a well-established tool for solving it. In this way, we transform the heuristic selection of SAE features and manual strength adjustment into an optimization process that is more interpretable, automated, and stable in practice. We believe that this simple yet effective combination is also valuable, particularly in a new and practically meaningful application scenario.
>
> ### (Q3) Sensitivity to sparsity budget
>
> | Support Range | Alignment | Consistency | Quality | Mean Harmonic |
> | --- | --- | --- | --- | --- |
> | [5, 10] | 3.4 | 4.0 | 4.6 | 3.72 |
> | [5, 500] | 3.7 | 3.7 | 4.6 | 3.77 |
> | [2000, $\infty$] | 3.6 | 3.1 | 4.6 | 3.37 |
> | Unlimited | 3.7 | 3.7 | 4.6 | 3.77 |
>
> These results suggest that the default range [5, 500] acts more as a practical safeguard than a brittle hyperparameter. In our ablation, relaxing the bound gives similar performance to the default setting, while very small or very large supports degrade the overall trade-off. Therefore, in practice, it can also be left unconstrained and allowed to search over [0, $\infty$] without manual specification.

---

> > ### Author Rebuttal · Reviewer_Py6S · 2026-04-03
> >
> > Thank you for your rebuttal, which addressed my concerns. I would raise my score accordingly.

---

> > > ### Author Response · Authors · 2026-04-07
> > >
> > > Thank you very much for your encouraging follow-up and for confirming that our rebuttal addressed your concerns. We sincerely appreciate your time and consideration.
> > >
> > > In the revision, we will make the added ablations more explicit in the main text, clarify that the main gain primarily comes from optimization-based sparse support discovery (with ridge refit providing an additional improvement), and add the real-image editing validation as discussed.
> > >
> > > If any further clarification would be helpful for your final evaluation, we would be very happy to provide it.

---

### Official Review · Reviewer_mKbZ · 2026-03-12

**Soundness:** 3
**Presentation:** 3
**Significance:** 3
**Originality:** 3
**Overall Recommendation:** 4
**Confidence:** 2

**Summary:**

This work aims to replace heuristic, manual feature selection and strength tuning in SAE-based text-to-image editing with an optimization-based pipeline: steering-vector discovery is cast as a convex sparse recovery problem in the SAE decoder space, and steering strength is chosen by Bayesian optimization. The paper shows that SRLS improves the trade-off among alignment, consistency , and quality, using far fewer SAE features than previous works, and reports better or comparable LLM and objective metrics, including attribute removal and multi-attribute editing.

**Compliance With Llm Reviewing Policy:**

Affirmed.

**Final Justification:**

After reading the rebuttal, I will keep my original rating.

**Key Questions For Authors:**

- The following ablations would be helpful: (a) Lasso without ridge refit (use Lasso solution as (\Delta s) directly) vs. full SRLS, and (b) mean vs. max pooling for token aggregation, reporting harmonic mean (and optionally alignment/consistency/quality) under the same evaluation protocol? This would help separate the benefit of “sparse support” from “debiasing” and from “aggregation.”

- Support range ([k_{\min}, k_{\max}] = [5, 500]): How was this range chosen (pilot experiments, SAE size, or otherwise)? Have you tried other ranges (e.g., [10, 200] or [50, 1000]) and, if so, how stable are the main conclusions (e.g., Table 1 and Figure 4)?

- The main LLM results report each method at the γ that maximizes the harmonic mean per prompt–attribute–token. Could you clarify in the main text that this is “best strength per combination” and briefly discuss whether you expect the relative ranking of SRLS vs. SAEdit to change under a fixed-γ protocol (e.g., at γ = 5 or 7)? Appendix B already gives fixed-γ curves. A sentence in Section 6.2 or 6.3 linking the two would suffice.

- Appendix I mentions dependence on SAE capacity and issues with extreme attributes. Can you give one or two concrete failure examples (e.g., attribute + prompt where SRLS clearly fails or degrades consistency), and whether you observed systematic failure patterns (e.g., certain attribute types or subject categories)?

**Strengths And Weaknesses:**

**Strengths**

- The motivation (alignment vs. consistency vs. quality) is clear. Using reconstruction-space alignment (Eq. 5–6) and the affine decoder to obtain a single mean target (\bar{d}) and a Lasso objective is a clean and well-explained idea. The argument that aligning in decoder space avoids unattainable directions and yields more stable, sparse solutions is plausible and consistent with the experiments.

**Weaknesses**

- There is no systematic ablation for: (1) Lasso-only vs. Lasso + ridge refit (contribution of debiasing); (2) aggregation (mean vs. max pooling) for prompt-level (\bar{s}^+, \bar{s}^-); (3) λ-path and support range ([k_{\min}, k_{\max}] = [5, 500]). How this range was chosen and how sensitive results are to it. Table 2 ablates BO vs. grid for γ but not the refit or aggregation. Without these, it is harder to attribute gains to “sparse recovery + debiasing” vs. other design elements.

- For each prompt–attribute–token combination, the best γ (maximizing harmonic mean of the three LLM scores) is used as the representative result. This “best strength per combination” favors methods that have a good peak somewhere in γ. The paper does provide fixed-γ curves (Appendix B, Figure 10), which is helpful, but the main tables (e.g., Figure 4, Table 2) are at “best γ.” Clarifying that the primary comparison is at “oracle” strength and discussing how this might affect conclusions would improve transparency. For SEGA and Pix2Pix-Zero, strength is not directly comparable; the text correctly notes that SEGA often under-edits (high consistency, lower alignment).

- Mean support size (~307) is an outcome of the λ-path over [5, 500] and the data. Sensitivity to ([k_{\min}, k_{\max}]) and to the number/quality of contrastive prompt pairs (14 from SAEdit + 19 from Appendix J templates) is not reported. If performance or sparsity is sensitive to these, it would be useful to state so and, if possible, give a brief sensitivity analysis.

- The paper acknowledges that at high γ, SRLS can lose consistency due to the exponential schedule and clipping (Section 6.3). A short discussion of recommended γ ranges or of a more robust schedule could help practitioners.

- There are no guarantees on support recovery or estimation error for the Lasso (Eq. 8) in this setting (e.g., restricted eigenvalue or compatibility conditions on (W_d)). The design matrix is determined by the SAE decoder. A brief comment on when the reduction is expected to work well (e.g., column coherence of (W_d)) would strengthen the foundation.

---

> ### Author Rebuttal · Authors · 2026-03-31
>
> We thank the reviewer for the careful and detailed feedback. We fully agree that these questions are important. To meet the rebuttal deadline, we added a small-scale ablation study (using the same setting of $\gamma\in\\{1,2,...,7\\}$ throughout):
>
> ### (W1 & Q1 & Q2) Systematic ablations
>
> | Method | Alignment | Consistency | Quality | Mean Harmonic |
> | --- | --- | --- | --- | --- |
> | All | 2.9 | 3.1 | 4.6 | 3.08 |
> | Lasso | 3.6 | 3.7 | 4.6 | 3.71 |
> | Lasso + Ridge (Max) | 3.5 | 3.6 | 4.5 | 3.65 |
> | Lasso + Ridge (Mean) | 3.7 | 3.7 | 4.6 | 3.77 |
>
> For the pooling choice, mean pooling is more stable for learning the “global attribute shift,” so we use mean pooling as the default aggregation method.
>
> For Lasso-only vs. Lasso + ridge refit, the current ablation shows that Lasso-only achieves a Mean Harmonic of 3.71, while Lasso + Ridge (Mean) achieves 3.77. This suggests that sparse support discovery already brings a clear gain over All, and ridge refit provides a further small improvement.
>
> | Support Range | Alignment | Consistency | Quality | Mean Harmonic |
> | --- | --- | --- | --- | --- |
> | [5, 10] | 3.4 | 4.0 | 4.6 | 3.72 |
> | [5, 500] | 3.7 | 3.7 | 4.6 | 3.77 |
> | [2000, $\infty$] | 3.6 | 3.1 | 4.6 | 3.37 |
> | Unlimited | 3.7 | 3.7 | 4.6 | 3.77 |
>
> The support-range ablation shows that too small a support leads to under-editing, while very large supports degrade the alignment/consistency trade-off and reduce the overall harmonic mean. In fact, our method can automatically select an appropriate support size (the average number of selected features is around 310 for both [5, 500] and Unlimited), so the default range [5, 500] acts as a practical safeguard than a brittle hyperparameter. In practice, we can simply leave it unconstrained and let the method search automatically.
>
> ### (W2 & Q3) Performance under fixed $\gamma$
>
> This is a very helpful point. We will clarify in the main paper that the main results in Figure 4 / Table 2 are based on an **oracle-style comparison using the best strength for each prompt–attribute–token combination**. We use this protocol because different SAE-based methods have different numbers of features and different activation scales, so directly fixing $\gamma$ is not fully fair.
>
> The fixed-$\gamma$ curves in the appendix B suggest that SRLS remains stronger than SAEdit and All across a broad range of strengths. Thus, the qualitative conclusion does not change under a fixed-$\gamma$ protocol, although the margin may differ. Finally, according to the current curves, it seems that a $\gamma$ in the range of 1–4 would be a reasonable default choice that achieves a good trade-off, but we recommand to use BO to select the best strength for each case if possible.
>
> ### (W3) Theoretical guarantees
>
> This paper does not prove support recovery or estimation error guarantees. The current theoretical contribution is mainly that, with the affine decoder, the problem can be reduced exactly to a mean-target Lasso problem. Classical Lasso recovery guarantees would further require additional conditions on \(W_d\), such as low column correlation and good compatibility / restricted-eigenvalue properties. We will add this discussion in the revised paper.
>
> ### (Q4) Failure cases
>
> We will add more concrete failure modes in the revised paper. At present, we have not found a fully systematic failure pattern, but we do observe that some extreme attributes (e.g., “crying”) lead to lower edit quality. A possible reason is that the model itself has a weaker understanding of these attributes (indeed, even direct prompting does not generate high-quality crying images).

---

> > ### Author Rebuttal · Reviewer_mKbZ · 2026-04-03
> >
> > Thank the author for providing the additional data and explanation. I am leaning towards keeping my original rating.

---

> > > ### Author Response · Authors · 2026-04-07
> > >
> > > Thank you again for the careful follow-up and for acknowledging the additional data and explanations..
> > >
> > > In the revision, we will explicitly strengthen the paper along the directions you highlighted: clearer presentation of the Lasso / ridge-refit / pooling ablations, clearer wording that the main comparison is under an oracle-style best-strength protocol together with discussion of the fixed-γ results, added failure-case analysis, and a more explicit discussion of when the Lasso reduction is expected to work well.
> > >
> > > We sincerely appreciate these suggestions, and they will directly improve the final version of the paper.

---

### Official Review · Reviewer_81XN · 2026-03-21

**Soundness:** 4
**Presentation:** 3
**Significance:** 3
**Originality:** 3
**Overall Recommendation:** 4
**Confidence:** 4

**Summary:**

This paper proposes Sparse Relaxed-Lasso Steering (SRLS) for text-guided image editing with frozen diffusion models and a pretrained SAE on the text encoder. The key idea is to learn a sparse universal steering direction in SAE feature space from contrastive prompt pairs. Using the affine SAE decoder, the authors reduce steering discovery to a Lasso objective, then perform support-restricted ridge refitting to debias coefficients and Bayesian optimization to tune the inference-time steering strength. The goal is to improve the trade-off between edit alignment, content preservation, and image quality compared with prior training-free editing methods and heuristic SAE-based baselines.

**Compliance With Llm Reviewing Policy:**

Affirmed.

**Final Justification:**

The authors addressed all my concerns, I'd like to maintain my score.

**Key Questions For Authors:**

1. How are the contrastive prompt pairs constructed, and how many are needed per concept before performance saturates?
2. Are baselines given a comparable tuning budget for steering strength, so that the comparison is fair?
3. How sensitive is the method to the support-size range, pooling choice, and the reward function used for optimizing gamma?
4. How well does the learned direction generalize to prompts that differ substantially from the prompt pairs used to fit it?

**Limitations:**

Yes

**Strengths And Weaknesses:**

## Strengths

- The paper presents a clear and principled formulation. Replacing heuristic feature selection with a sparse Lasso optimization objective is the main strength.
- The reduction to a convex Lasso problem in decoder space is elegant and technically clean. It is simple yet effective. The combination of Lasso for support selection and ridge refitting for debiasing is sensible and well motivated.
- The method is appealing from an interpretability perspective, since it stresses a sparser concept direction in SAE feature space.
- The work addresses a real weakness of prior training-free editing methods, namely unstable feature selection and sensitivity to steering strength.

## Weaknesses
- The assumption that a single universal sparse direction can capture an attribute across many prompt contexts may be too strong, especially for context-dependent or multimodal concepts.
- The prompt-level mean pooling may remove useful token-level information and could limit more localized or syntax-sensitive edits.
- It is unclear how much of the final gain comes from the core sparse direction discovery versus the additional per-instance search over steering strength.

---

> ### Author Rebuttal · Authors · 2026-03-31
>
> We thank the reviewer for the careful reading and positive feedback. For faster turnaround, the additional rebuttal ablations below use a reduced sweep, $\gamma \in \\{1,2,\ldots,7\\}$, to check trends rather than replace the main benchmark numbers.
>
>
> ### (W1) The assumption of a single universal sparse direction
> We agree that this assumption does not hold perfectly for all attributes. Our claim is more limited: for a common class of attribute edits, there exists a stable average semantic shift across multiple contrastive prompt pairs, and this shift can be well approximated by a sparse direction in the SAE reconstruction space. Our experiments suggest that the “universal direction” is empirically effective in our setting. We will clarify this point in the revision and state its scope more carefully.
>
> ### (W2) Mean pooling v.s. Max pooling
> Mean pooling is used only during direction learning to build a stable prompt-level target. At inference, the learned direction is still applied token-wise, either to all tokens or only to selected subject tokens. We agree that this may trade off some syntax sensitivity for robustness; empirically, mean pooling performs better than max pooling in our rebuttal ablation as shown below:
>
> | Pooling | Alignment | Consistency | Quality | Mean Harmonic |
> | --- | --- | --- | --- | --- |
> | Mean | 3.7 | 3.7 | 4.6 | **3.77** |
> | Max | 3.5 | 3.6 | 4.5 | 3.65 |
>
> ### (W3) The gains of sparse direction discovery and BO search
> The gains are not solely due to BO. Even **without BO** (Fig. 4), SRLS already outperforms the SAE baselines: HM 3.97 vs. 3.89 for SAEdit and 3.32 for **All** (the most basic baseline). Relative to **All**, sparse direction discovery improves HM by 0.65. BO further improves all SAE-based methods under the same evaluation budget, while SRLS remains best or tied-best (Table 2). Thus, sparse direction discovery already provides a clear advantage, and BO is an orthogonal improvement.
>
> ### (Q1) Performance saturation with contrastive prompt pairs
> For the original attribute set, we use the positive/negative prompt pairs from SAEdit; for the additional complex attributes, we generate contrastive pairs using the prompt templates in Appendix J. Each attribute contains 50–150 prompt pairs.
>
> Under the same setting, we compared performance when learning the steering direction from different numbers of prompt pairs:
>
> | #pair | Alignment | Consistency | Quality | Mean Harmonic |
> | --- | --- | --- | --- | --- |
> | 25 | 3.5 | 3.7 | 4.6 | 3.70 |
> | 50 | 3.7 | 3.7 | 4.6 | 3.77 |
> | 100 | 3.6 | 3.8 | 4.6 | 3.77 |
>
> As shown above, 50-100 pairs per attribute are already sufficient to learn a direction that generalizes well.
>
>
> ### (Q2) Comparable tuning budget
>
> Yes. We state this in Section 6.1. For all SAE-based methods (All, SAEdit, and Ours), we ensure a fair tuning budget:
> - all steering vectors are normalized to unit $\ell_2$ norm;
> - grid search uses the same 21 values of $\gamma$;
> - BO also uses the same budget of 21 evaluations over the same search range;
> - each candidate $\gamma$ is evaluated 3 times independently, and we report the average to reduce noise.
>
> Therefore, the strength-tuning budget is aligned across all SAE-based methods. For non-SAE methods such as SEGA and Pix2Pix-Zero, we use their recommended hyperparameters.
>
> ### (Q3) Sensitivity to support-size range, pooling choice, and reward function for optimizing gamma
>
> For pooling, we have already discussed the advantage of mean pooling above.
>
> For the support size, our method can automatically choose an appropriate support size; the range used in the experiments is only a safe bound. In fact, removing the bound in this ablation gives similar performance to the default setting, while forcing very large supports degrades performance.
>
> | Support Range | Alignment | Consistency | Quality | Mean Harmonic |
> | --- | --- | --- | --- | --- |
> | [5, 10] | 3.4 | 4.0 | 4.6 | 3.72 |
> | [5, 500] | 3.7 | 3.7 | 4.6 | 3.77 |
> | [2000, $\infty$] | 3.6 | 3.1 | 4.6 | 3.37 |
> | Unlimited | 3.7 | 3.7 | 4.6 | 3.77 |
>
> This is consistent with the trend reported in the paper: a support set that is too small leads to under-editing, while a support set that is too large may hurts alignment and consistency.
>
> Different reward functions lead BO to prefer different $\gamma$, trading off alignment, consistency, and quality. We will clarify this and add corresponding results in the revision.
>
> ### (Q4) Generalization to prompts that differ substantially from the fitting prompt pairs
>
> We provide encouraging evidence of generalization within the studied setting: evaluation on 17 unseen base prompts, four subject categories, reuse of the same direction across categories (e.g., *sunglasses* in Fig. 7), attribute erasing, multi-attribute composition, and consistent trends on Stable Diffusion 3.5 Large. We will make this scope more explicit in the revision.

---

> > ### Author Rebuttal · Reviewer_81XN · 2026-04-06
> >
> > Good paper, I will maintain my score.

---

> > > ### Author Response · Authors · 2026-04-07
> > >
> > > Thank you very much for the positive follow-up and for the careful evaluation of our paper. We are glad that our rebuttal addressed your concerns.
> > >
> > > In the revision, we will incorporate the added clarifications and ablations more clearly in the main text, including the discussion of prompt-pair quantity, pooling choice, support-size sensitivity, and the role of Bayesian optimization.
> > >
> > > We sincerely appreciate your time and feedback.

---

### Decision · Program_Chairs · 2026-04-30

**Decision:**

Accept (regular)

**Comment:**

The paper presents a clear and technically sound formulation for SAE-based text-guided image editing, with an elegant reduction of steering-vector discovery to a sparse optimization problem and generally solid empirical results.

However, the discussion converged on several concerns that ultimately limit enthusiasm: the methodological novelty appears somewhat incremental, as the approach mainly combines existing tools such as Lasso, ridge refit, and Bayesian optimization in the steering setting; the evaluation remains somewhat limited, especially without comprehensive validation on real-image editing and more challenging binding/compositional cases; and some practical aspects, including sensitivity to hyperparameters, dependence on multiple paired prompts per attribute, and the role of tuning in the reported gains, still weaken the overall impact. While the rebuttal addressed several reviewer questions and improved the quality of the paper, concerns remain about originality, breadth of validation, and practical significance.